

# Ellipsoids (v1.0): 3D Magnetic modelling of ellipsoidal bodies

Diego Takahashi Tomazella[1] and Vanderlei C. Oliveira Jr.[1]

[1]Department of Geophysics, Observatório Nacional, Rio de Janeiro, Brazil

*Correspondence to:* Vanderlei C. Oliveira Jr. (vandscoelho@gmail.com)

**Abstract.**

A considerable amount of literature has been published on the magnetic modelling of uniformly magnetized ellipsoids since the second half of the nineteenth century. Ellipsoids have flexibility to represent a wide range of geometrical forms, are the only known bodies which can be uniformly magnetized in the presence of a uniform inducing field and are the only bodies
for which the self-demagnetization can be treated analytically. This property makes ellipsoids particularly useful for modelling compact orebodies having high susceptibility. In this case, neglecting the self-demagnetization may strongly mislead the interpretation of these bodies by using magnetic methods. A number of previous studies consider that the self-demagnetization can be neglected for the case in which the geological body has an isotropic susceptibility lower than or equal to 0.1 SI. This limiting value, however, seems to be determined empirically and there has been no discussion about how this value was determined. Be-
sides, the geoscientific community lacks an easy-to-use tool to simulate the magnetic field produced by uniformly magnetized ellipsoids. Here, we present an integrated review of the magnetic modelling of arbitrarily oriented triaxial, prolate and oblate ellipsoids. Our review includes ellipsoids with both induced and remanent magnetization, as well as with isotropic or anisotropic susceptibility. We also propose a way of determining the isotropic susceptibility above which the self-demagnetization must be taken into consideration. Tests with synthetic data validate our approach. Finally, we provide a set of routines to model the
magnetic field produced by ellipsoids. The routines are written in Python language as part of the Fatiando a Terra, which is an open-source library for modelling and inversion in geophysics.

## 1 Introduction

Based on the mathematical theory of the magnetic induction developed by Poisson (1824), Maxwell (1873) affirmed that, if $U$ is the gravitational potential produced by any body with uniform density $\rho$ and arbitrary shape at a point $(x, y, z)$, then $-\frac{\partial U}{\partial x}$ is
the magnetic scalar potential produced at the same point by the same body if it has a uniform magnetization oriented along $x$ with intensity $\rho$. Maxwell (1873) generalized this idea as a way of determining the magnetic scalar potential produced by any uniformly magnetized body in a given direction. By presuming that this uniform magnetization is due to induction and that it is proportional to the resulting magnetic field (intensity) inside the body, he postulated that the resulting field must also be uniform and parallel to the magnetization. This uniformity is due to the fact that the resulting field is defined as the negative
gradient of the magnetic scalar potential. As a consequence of this uniformity, the gravitational potential $U$ at points within the body must be a quadratic function of the spatial coordinates. Apparently, Maxwell (1873) was the first one to postulate that





ellipsoids are the only finite bodies having a gravitational potential which satisfies this property and hence can be uniformly magnetized in the presence of a uniform inducing magnetic field. This property can be extended to other bodies defined as limiting cases of an ellipsoid (e.g., spheres, elliptic cylinders), however all the remaining non-ellipsoidal bodies cannot be uniformly magnetized in the presence of a uniform inducing field.

Another particularity of ellipsoids is that they are the only bodies which enable an analytical computation of its self-demagnetization. The self-demagnetization contributes to decrease the magnitude of the magnetization along the shortest axes of a body. It is a function of the body shape and gives rise to shape anisotropy (Uyeda et al., 1963; Thompson and Oldfield, 1986; Dunlop and Özdemir, 1997; Clark and Emerson, 1999; Tauxe, 2003). It is well-established in the literature that the self-demagnetization can be neglected if the body has a susceptibility lower than 0.1 SI (Emerson et al., 1985; Clark et al., 1986;

Eskola and Tervo, 1980; Guo et al., 1998, 2001; Purss and Cull, 2005; Hillan and Foss, 2013; Austin et al., 2014; Clark, 2014). On the other hand, neglecting the self-demagnetization in geological bodies with high susceptibilities (> 0.1 SI) may strongly mislead the interpretation obtained from magnetic methods. This limiting value, however, seems to be determined empirically and, so far, there has been little discussion about how it was determined.

Farrar (1979) demonstrated the importance of the ellipsoidal model in taking into account the self-demagnetization and

determining reliable drilling directions on the Tennant Creek field, Australia. Posteriorly, Hoschke (1991) also showed how the ellipsoidal model proved to be highly successful in locating and defining ironstone bodies in the Tennant Creek field. Clark (2000) provides a good discussion about the influence of the self-demagnetization in magnetic interpretation of the Osborne copper-gold deposit, Australia. This deposit is hosted by ironstone bodies that have very high susceptibility. According to Clark (2000), neglecting the effects of self-demagnetisation led errors of $\approx 55°$ in the interpreted dip. Recently, Austin

et al. (2014) used magnetic modelling and rock property measurements to show that, contrary to previous interpretations, the magnetization of the Candelaria iron oxide copper-gold deposit, Chile, is not dominated by the induced component. Rather, the deposit has a relatively weak remanent magnetization and is strongly affected by self-demagnetization. These examples show the importance of the self-demagnetization and the ellipsoidal model in producing trustworthy geological models of high-susceptibility orebodies, which may save significant cost associated with drilling.

A vast literature about the magnetic modelling of ellipsoidal bodies was developed in which are to be found the names of many researchers. Nevertheless, interest in this subject has not yet died out, as is evidenced by a list of modern papers in this field. Besides, the geoscientific community lacks an easy-to-use tool to simulate the magnetic field produced by uniformly magnetized ellipsoids. Such a tool could prove to be useful either for teaching and researching geophysics.

In this work, we present a review of the vast literature about the magnetic modelling of ellipsoidal bodies and a theoretical

discussion about the determination of the isotropic susceptibility value above which the self-demagnetization must be taken into consideration. We propose an alternative way of determining this value based on the body shape and the maximum relative error allowed in the resultant magnetization. This alternative approach is validated by the results obtained with numerical simulations. We also provide a set of routines to model the magnetic field produced by ellipsoids. The routines are written in Python language as part of the Fatiando a Terra (Uieda et al., 2013), which is an open-source library for modelling and inversion





in geophysics. We attempt to use the best practices of continuous integration, documentation, unit-testing, and version-control for the purpose of providing a reliable and easy-to-use code.

## 2 Methodology

### 2.1 Geometrical parameters and coordinate systems

Let $(x, y, z)$ be a point referred to a Cartesian coordinate system with axes $x$, $y$ and $z$ pointing to, respectively, North, East and down. For convenience, we denominate this coordinate system as *main coordinate system* (Fig. 1a). Let us consider an ellipsoidal body with centre at the point $(x_c, y_c, z_c)$, orientation defined by the angles *strike $\varepsilon$*, *dip $\zeta$*, and *rake $\eta$* (Fig. 1a), and semi-axes defined by positive constants $a$, $b$, $c$ (Fig. 1b). The orientation angles *strike*, *dip*, and *rake* are commonly used to define the orientation of lines in structural geology (Pollard and Fletcher, 2005; Allmendinger et al., 2012). The points $(x, y, z)$

located on the surface of this ellipsoidal body satisfy the following equation:

$$(\mathbf{r} - \mathbf{r}_c)^T \mathbf{A} (\mathbf{r} - \mathbf{r}_c) = 1, \tag{1}$$

where $\mathbf{r} = \begin{bmatrix} x & y & z \end{bmatrix}^\top$, $\mathbf{r}_c = \begin{bmatrix} x_c & y_c & z_c \end{bmatrix}^\top$, $\mathbf{A}$ is a positive definite matrix given by

$$\mathbf{A} = \mathbf{V} \begin{bmatrix} a^{-2} & 0 & 0 \\ 0 & b^{-2} & 0 \\ 0 & 0 & c^{-2} \end{bmatrix} \mathbf{V}^\top, \tag{2}$$

and $\mathbf{V}$ is an orthogonal matrix whose first, second and third columns are defined by unit vectors $\mathbf{v}_1$, $\mathbf{v}_2$, and $\mathbf{v}_3$ (Fig. 1b),

respectively. The matrix $\mathbf{V}$ can be defined in terms of three rotation matrices:

$$\mathbf{R}_1(\theta) = \begin{bmatrix} 1 & 0 & 0 \\ 0 & \cos\theta & \sin\theta \\ 0 & -\sin\theta & \cos\theta \end{bmatrix}, \tag{3}$$

$$\mathbf{R}_2(\theta) = \begin{bmatrix} \cos\theta & 0 & -\sin\theta \\ 0 & 1 & 0 \\ \sin\theta & 0 & \cos\theta \end{bmatrix} \tag{4}$$

and

$$\mathbf{R}_3(\theta) = \begin{bmatrix} \cos\theta & \sin\theta & 0 \\ -\sin\theta & \cos\theta & 0 \\ 0 & 0 & 1 \end{bmatrix}. \tag{5}$$

For triaxial ellipsoids (i.e., $a > b > c$) and prolate ellipsoids (i.e., $a > b = c$), we define the orthogonal matrix $\mathbf{V}$ as follows:

$$\mathbf{V} = \mathbf{R}_1\left(\frac{\pi}{2}\right) \mathbf{R}_2(\varepsilon) \mathbf{R}_1\left(\frac{\pi}{2} - \zeta\right) \mathbf{R}_3(\eta). \tag{6}$$



For oblate ellipsoids (i.e., $a < b = c$), we define $\mathbf{V}$ as follows:

$$\mathbf{V} = \mathbf{R}_3\left(-\frac{\pi}{2}\right) \mathbf{R}_1\left(\pi\right) \mathbf{R}_3\left(\varepsilon\right) \mathbf{R}_2\left(\frac{\pi}{2} - \zeta\right) \mathbf{R}_1\left(\eta\right) . \tag{7}$$

The orthogonal matrices $\mathbf{V}$ used here for triaxial, prolate and oblate ellipsoids (Eqs. 6 and 7) are different from that used by Emerson et al. (1985) and Clark et al. (1986).

The magnetic modelling of an ellipsoidal body is commonly performed in a particular Cartesian coordinate system that is aligned with the body semi-axes and has the origin coincident with the body centre (Fig. 1b). For convenience, we denominate this particular coordinate system as *local coordinate system*. The relationship between the Cartesian coordinates $(\tilde{x}, \tilde{y}, \tilde{z})$ of a point in a local coordinate system and the Cartesian coordinates $(x, y, z)$ of the same point in the main system is given by:

$$\tilde{\mathbf{r}} = \mathbf{V}^{\top}\left(\mathbf{r} - \mathbf{r}_c\right) , \tag{8}$$

where $\tilde{\mathbf{r}} = \begin{bmatrix} \tilde{x} & \tilde{y} & \tilde{z} \end{bmatrix}^{\top}$, $\mathbf{r}$ and $\mathbf{r}_c$ are defined in Eq. 1 and the matrix $\mathbf{V}$ (Eq. 2) is defined according to the ellipsoid type. Subsequently, quantities referred to the local coordinate system (Fig. 1b) are represented with the simbol "$\sim$".

## 2.2 Theoretical background

Consider a magnetized ellipsoid immersed in a uniform inducing magnetic field $\mathbf{H}_0$ (in $\mathrm{Am}^{-1}$) given by

$$\mathbf{H}_0 = \|\mathbf{H}_0\| \begin{bmatrix} \cos I \cos D \\ \cos I \sin D \\ \sin I \end{bmatrix} , \tag{9}$$

where $\|\cdot\|$ denotes the Euclidean norm (or 2-norm) and $D$ and $I$ are respectively, the declination and inclination of the local-geomagnetic field in the main coordinate system (Fig. 1a). This field represents the main component of the Earth's magnetic field, which is usually assumed to be generated by the Earth's liquid core. In the absence of conduction currents, the total magnetic field $\mathbf{H}(\mathbf{r})$ at the position $\mathbf{r}$ (Eq. 1) of a point referred to the main coordinate system is defined as follows (Sharma, 1966; Eskola and Tervo, 1980; Reitz et al., 1992; Stratton, 2007):

$$\mathbf{H}(\mathbf{r}) = \mathbf{H}_0 - \nabla V(\mathbf{r}) , \tag{10}$$

where the second term is the negative gradient of the magnetic scalar potential $V(\mathbf{r})$ given by:

$$V(\mathbf{r}) = -\frac{1}{4\pi} \iiint\limits_{\vartheta} \mathbf{M}(\mathbf{r}')^{\top} \nabla \left(\frac{1}{\|\mathbf{r} - \mathbf{r}'\|}\right) dx' dy' dz' . \tag{11}$$

In this equation, $\mathbf{r}' = \begin{bmatrix} x' & y' & z' \end{bmatrix}^{\top}$ is the position vector of a point located within the volume $\vartheta$, the integral is conducted over the variables $x'$, $y'$ and, $z'$ and $\mathbf{M}(\mathbf{r}')$ is the magnetization vector (in $\mathrm{Am}^{-1}$). Eq. 11 is valid anywhere, independently 25    if the position vector $\mathbf{r}$ represents a point located inside or outside the magnetized body (DuBois, 1896; Reitz et al., 1992; Stratton, 2007).





Based on Maxwell's postulate, let us assume that the body has a uniform magnetization given by

$$\mathbf{M} = \mathbf{K}\,\mathbf{H}^{\dagger}\,, \tag{12}$$

where $\mathbf{H}^{\dagger}$ is the resultant uniform magnetic field at any point within the body and $\mathbf{K}$ is a constant and symmetrical 2nd-order tensor representing the magnetic susceptibility of the body. This is a good approximation for bodies at room temperature, subjected to an inducing field $\mathbf{H}_0$ with strength $\leq 10^{-3}\mu_0^{-1}$ A m$^{-1}$ (Rochette et al., 1992), where $\mu_0$ represents the magnetic constant (in H m$^{-1}$). In this case, the susceptibility tensor $\mathbf{K}$ is commonly represented, in the main coordinate system (Fig. 1a), as follows:

$$\mathbf{K} = \mathbf{U} \begin{bmatrix} k_1 & 0 & 0 \\ 0 & k_2 & 0 \\ 0 & 0 & k_3 \end{bmatrix} \mathbf{U}^{\top}\,, \tag{13}$$

where $k_1 > k_2 > k_3$ are the *principal susceptibilities* and $\mathbf{U}$ is an orthogonal matrix whose columns $\mathbf{u}_i$, $i = 1,2,3$, are unit vectors called *principal directions*. Similarly to the matrix $\mathbf{V}$ (Eqs. 1, 6 and 7), we define the matrix $\mathbf{U}$ as a function of given orientation angles $\varepsilon$, $\zeta$, and $\eta$ depending on the ellipsoid type. For triaxial and prolate ellipsoids, we define $\mathbf{U}$ by using Eq. 6, whereas for oblate ellipsoids we use Eq. 7. Notice that the orientation angles $\varepsilon$, $\zeta$, and $\eta$ defining the orthogonal matrix $\mathbf{U}$ may be different from that angles $\varepsilon$, $\zeta$, and $\eta$ defining the ellipsoid orientation (Fig. 1).

If the principal susceptibilities are different from each other, we say that the body has an anisotropy of magnetic susceptibility (AMS). The AMS is generally associated to the preferred orientation of the grains of magnetic minerals forming the rock (Fuller, 1963; Uyeda et al., 1963; Janák, 1972; Hrouda, 1982; Thompson and Oldfield, 1986; MacDonald and Ellwood, 1987; Rochette et al., 1992; Dunlop and Özdemir, 1997; Tauxe, 2003). For the particular case in which the principal directions coincide with the ellipsoid axes, the matrix $\mathbf{U}$ is equal to the matrix $\mathbf{V}$ (Eq. 2). Another important particular case is that in which the susceptibility is isotropic and, consequently, the principal susceptibilities $k_1$, $k_2$, and $k_3$ (Eq. 13) are equal to a constant $\chi$. In this case, the susceptibility tensor $\mathbf{K}$ (Eq. 13) assumes the particular form

$$\mathbf{K} = \chi\,\mathbf{I}\,, \tag{14}$$

where $\mathbf{I}$ represents the identity matrix.

By using the magnetization $\mathbf{M}$ defined by Eq. 12, the total magnetic field $\mathbf{H}(\mathbf{r})$ (Eq. 10) can be rewritten as follows:

$$\mathbf{H}(\mathbf{r}) = \mathbf{H}_0 + \mathbf{N}(\mathbf{r})\,\mathbf{K}\,\mathbf{H}^{\dagger}\,, \tag{15}$$

where $\mathbf{N}(\mathbf{r})$ is a symmetrical matrix whose $ij$-element $n_{ij}(\mathbf{r})$ is given by

$$n_{ij}(\mathbf{r}) = \frac{1}{4\pi}\frac{\partial^2 f(\mathbf{r})}{\partial r_i \partial r_j}\,, \quad i = 1,2,3\,, \quad j = 1,2,3\,, \tag{16}$$

$r_1 = x$, $r_2 = y$, $r_3 = z$ are the elements of the position vector $\mathbf{r}$ (Eq. 1), and

$$f(\mathbf{r}) = \iiint\limits_{\vartheta} \frac{1}{\|\mathbf{r} - \mathbf{r}'\|}\,dx'dy'dz'\,. \tag{17}$$



Notice that the scalar function $f(\mathbf{r})$ (Eq. 17) is proportional to the gravitational potential that would be produced by the ellipsoidal body with volume $\vartheta$ if it had a uniform density equal to the inverse of the gravitational constant. It can be shown that the elements $n_{ij}(\mathbf{r})$ are finite whether $\mathbf{r}$ is a point within or without the volume $\vartheta$ (Peirce, 1902; Webster, 1904). The matrix $\mathbf{N}(\mathbf{r})$ (Eq. 15) is called *depolarization tensor* (Solivérez, 1981, 2008).

The following part of this paper moves on to describe the magnetic field $\mathbf{H}(\mathbf{r})$ (Eq. 15) at points located both within and without the volume $\vartheta$ of the ellipsoidal body. However, the mathematical developments are conveniently performed in the local coordinate system (Fig. 1b) related to the respective ellipsoidal body.

## 2.3    Coordinate transformation

To continue our description of the magnetic modelling of ellipsoidal bodies, it is convenient to perform two important coordi-
nate transformations. The first one transforms the scalar function $f(\mathbf{r})$ (Eq. 17) from the main coordinate system (Fig. 1a) into a new scalar function $\tilde{f}(\tilde{\mathbf{r}})$ referred to the local coordinate system (Fig. 1b). The function $\tilde{f}(\tilde{\mathbf{r}})$ was first presented by Dirichlet (1839) to describe the gravitational potential produced by homogeneous ellipsoids. Posteriorly, several authors also deduced and used this function for describing the magnetic and gravitational fields produced by triaxial, prolate, and oblate ellipsoids (Maxwell, 1873; Thomson and Tait, 1879; DuBois, 1896; Peirce, 1902; Webster, 1904; Kellogg, 1929; Stoner, 1945; Osborn,
1945; Peake and Davy, 1953; Macmillan, 1958; Chang, 1961; Lowes, 1974; Clark et al., 1986; Tejedor et al., 1995; Stratton, 2007).

It is convenient to use $\tilde{f}^{\dagger}(\tilde{\mathbf{r}})$ and $\tilde{f}^{\ddagger}(\tilde{\mathbf{r}})$ to define the function $\tilde{f}(\tilde{\mathbf{r}})$ evaluated, respectively, at points $\tilde{\mathbf{r}}$ inside and outside the volume $\vartheta$ of the ellipsoidal body. The scalar function $\tilde{f}^{\dagger}(\tilde{\mathbf{r}})$ is given by

$$\tilde{f}^{\dagger}(\tilde{\mathbf{r}}) = \pi\,abc \int\limits_{0}^{\infty} \left( 1 - \frac{\tilde{x}^2}{a^2+u} - \frac{\tilde{y}^2}{b^2+u} - \frac{\tilde{z}^2}{c^2+u} \right) \frac{1}{R(u)}\,du\,, \quad \tilde{\mathbf{r}} \in \vartheta\,, \tag{18}$$

where

$$R(u) = \sqrt{(a^2+u)\,(b^2+u)\,(c^2+u)}\,. \tag{19}$$

This function represents the gravitational potential that would be produced by the ellipsoidal body at points located within its volume $\vartheta$ if it had a uniform density equal to the inverse of the gravitational constant. Notice that, in this case, the gravitational potential is a quadratic function of the spatial coordinates $\tilde{x}$, $\tilde{y}$, and $\tilde{z}$, which supported the Maxwell's (1873) postulate about
uniformly magnetized ellipsoids. In a similar way, the function $\tilde{f}^{\ddagger}(\tilde{\mathbf{r}})$ is given by

$$\tilde{f}^{\ddagger}(\tilde{\mathbf{r}}) = \pi\,abc \int\limits_{\lambda}^{\infty} \left( 1 - \frac{\tilde{x}^2}{a^2+u} - \frac{\tilde{y}^2}{b^2+u} - \frac{\tilde{z}^2}{c^2+u} \right) \frac{1}{R(u)}\,du\,, \quad \tilde{\mathbf{r}} \notin \vartheta\,, \tag{20}$$

where $R(u)$ is defined by Eq. 19 and the parameter $\lambda$ is defined according to the ellipsoid type as a function of the spatial coordinates $\tilde{x}$, $\tilde{y}$, and $\tilde{z}$ (see Appendix B). For readers interested in additional information about the parameter $\lambda$, we recommend Webster (1904, p. 234), Kellogg (1929, p. 184) and Clark et al. (1986).



The second important coordinate transformation is defined with respect to Eq. 15. By properly using the orthogonality of matrix $\mathbf{V}$ (Eq. 2), the magnetic field $\mathbf{H}(\mathbf{r})$ (Eq. 15) can be transformed from the main coordinate system (Fig. 1a) to the local coordinate system (Fig. 1b) as follows:

$$\underbrace{\mathbf{V}^\top \mathbf{H}(\mathbf{r})}_{\tilde{\mathbf{H}}(\tilde{\mathbf{r}})} = \underbrace{\mathbf{V}^\top \mathbf{H}_0}_{\tilde{\mathbf{H}}_0} + \underbrace{\mathbf{V}^\top \mathbf{N}(\mathbf{r})\mathbf{V}}_{\tilde{\mathbf{N}}(\tilde{\mathbf{r}})} \underbrace{\mathbf{V}^\top \mathbf{K}\mathbf{V}}_{\tilde{\mathbf{K}}} \underbrace{\mathbf{V}^\top \mathbf{H}^\dagger}_{\tilde{\mathbf{H}}^\dagger}, \tag{21}$$

where the superscript "$\sim$" denotes quantities referred to the respective local coordinate system.

In Eq. 21, the transformed depolarization tensor $\tilde{\mathbf{N}}(\tilde{\mathbf{r}})$ is calculated as a function of the original depolarization tensor $\mathbf{N}(\mathbf{r})$ (Eq. 15). In this case, the elements of $\tilde{\mathbf{N}}(\tilde{\mathbf{r}})$ are calculated as a function of the second derivatives of the function $f(\mathbf{r})$ (Eq. 17), which is defined in the main coordinate system (Fig. 1a). It can be shown (Appendix A), however, that the elements $\tilde{n}_{ij}(\tilde{\mathbf{r}})$ of $\tilde{\mathbf{N}}(\tilde{\mathbf{r}})$ can also be calculated as follows:

$$\tilde{n}_{ij}(\tilde{\mathbf{r}}) = \frac{1}{4\pi} \frac{\partial^2 \tilde{f}(\tilde{\mathbf{r}})}{\partial \tilde{r}_i \partial \tilde{r}_j}, \quad i = 1, 2, 3, \quad j = 1, 2, 3, \tag{22}$$

where $\tilde{r}_1 = \tilde{x}$, $\tilde{r}_2 = \tilde{y}$, and $\tilde{r}_3 = \tilde{z}$ are the elements of the transformed vector $\tilde{\mathbf{r}}$ (Eq. 8) and $\tilde{f}(\tilde{\mathbf{r}})$ is given by Eq. 18 or 20, depending if $\tilde{\mathbf{r}}$ represents a point located within or without the volume $\vartheta$ of the ellipsoidal body.

### 2.4  Transformed depolarization tensors $\tilde{\mathbf{N}}(\tilde{\mathbf{r}})$

#### 2.4.1  Depolarization tensor $\tilde{\mathbf{N}}^\dagger$

Let $\tilde{\mathbf{N}}^\dagger$ be the transformed depolarization tensor calculated for the case in which $\tilde{\mathbf{r}}$ (Eq. 8) represents a point located outside the ellipsoidal body. In this case, the elements of $\tilde{\mathbf{N}}^\dagger$ are calculated according to Eq. 22, with $\tilde{f}(\tilde{\mathbf{r}})$ given by $\tilde{f}^\dagger(\tilde{\mathbf{r}})$ (Eq. 18). As we have already pointed out, the $\tilde{f}^\dagger(\tilde{\mathbf{r}})$ (Eq. 18) is a quadratic function of the spatial coordinates $\tilde{x}$, $\tilde{y}$ and $\tilde{z}$. Consequently, the elements $\tilde{n}_{ij}^\dagger$, $i = 1, 2, 3$, $j = 1, 2, 3$, of $\tilde{\mathbf{N}}^\dagger$ do not depend on the elements of the transformed position vector $\tilde{\mathbf{r}}$ (Eq. 8). Besides, the off-diagonal elements are zero and the diagonal elements are given by (Stoner, 1945):

$$\tilde{n}_{ii}^\dagger = \frac{abc}{2} \int\limits_0^\infty \frac{1}{(e_i^2 + u)\, R(u)}\, du, \quad i = 1, 2, 3, \tag{23}$$

where $R(u)$ is defined by Eq. 19 and $e_1 = a$, $e_2 = b$, and $e_3 = c$. These elements are commonly known as *demagnetizing factors* and are defined according to the ellipsoid type. Here, we calculate the demagnetizing factors in the SI system. Consequently, they satisfies the condition $\tilde{n}_{11}^\dagger + \tilde{n}_{22}^\dagger + \tilde{n}_{33}^\dagger = 1$, independently of the ellipsoid type. It is worth stressing that, according to Eq. 23, the demagnetizing factors $\tilde{n}_{ii}^\dagger$ are constants defined by the ellipsoid semi-axes $a$, $b$, and $c$.

Notice that, according to Eqs. 21 and A7,

$$\mathbf{N}(\mathbf{r}) = \mathbf{V}\, \tilde{\mathbf{N}}^\dagger\, \mathbf{V}^\top, \tag{24}$$

where $\tilde{\mathbf{N}}^\dagger$ is a diagonal matrix and $\mathbf{V}$ (Eq. 2) is an orthogonal matrix. This equation shows that, for the particular case in which $\mathbf{r}$ and consequently $\tilde{\mathbf{r}}$ represent a point inside the volume $\vartheta$ of the ellipsoid, the elements $\tilde{n}_{ii}^\dagger$ of $\tilde{\mathbf{N}}^\dagger$ represent the eigenvalues while the columns of $\mathbf{V}$ represent the eigenvectors of the original depolarization tensor $\mathbf{N}(\mathbf{r})$.





**Triaxial ellipsoids**

For triaxial ellipsoids (e.g., $a > b > c$), the demagnetizing factors obtained by solving Eq. 23 are given by:

$$\tilde{n}_{11}^{\dagger} = \frac{abc}{(a^2 - c^2)^{\frac{1}{2}} (a^2 - b^2)} \left[ F(\kappa, \phi) - E(\kappa, \phi) \right] , \tag{25}$$

$$\tilde{n}_{22}^{\dagger} = -\frac{abc}{(a^2 - c^2)^{\frac{1}{2}} (a^2 - b^2)} \left[ F(\kappa, \phi) - E(\kappa, \phi) \right] + \frac{abc}{(a^2 - c^2)^{\frac{1}{2}} (b^2 - c^2)} E(\kappa, \phi) - \frac{c^2}{b^2 - c^2} \tag{26}$$

and

$$\tilde{n}_{33}^{\dagger} = -\frac{abc}{(a^2 - c^2)^{\frac{1}{2}} (b^2 - c^2)} E(\kappa, \phi) + \frac{b^2}{b^2 - c^2} , \tag{27}$$

where

$$F(\kappa, \phi) = \int_0^{\phi} \frac{1}{\left(1 - \kappa^2 \sin^2 \psi\right)^{\frac{1}{2}}} d\psi , \tag{28}$$

and

$$E(\kappa, \phi) = \int_0^{\phi} \left(1 - \kappa^2 \sin^2 \psi\right)^{\frac{1}{2}} d\psi , \tag{29}$$

with $\kappa = \left[ \left(a^2 - b^2\right) / \left(a^2 - c^2\right) \right]^{\frac{1}{2}}$ and $\cos \phi = c/a$. The functions $F(\kappa, \phi)$ (Eq. 28) and $E(\kappa, \phi)$ (Eq. 29) are called Legendre's normal elliptic integrals of the first and second kind, respectively. Stoner (1945) presented a detailed deduction of the demagnetizing factors $\tilde{n}_{11}^{\dagger}$ (Eq. 25), $\tilde{n}_{22}^{\dagger}$ (Eq. 26) and $\tilde{n}_{33}^{\dagger}$ (Eq. 27). Clark et al. (1986) presented similar formulas. It can be shown that these demagnetizing factors satisfy the conditions $\tilde{n}_{11}^{\dagger} + \tilde{n}_{22}^{\dagger} + \tilde{n}_{33}^{\dagger} = 1$ and $\tilde{n}_{11}^{\dagger} < \tilde{n}_{22}^{\dagger} < \tilde{n}_{33}^{\dagger}$.

**Prolate ellipsoids**

For prolate ellipsoids (e.g., $a > b = c$), the demagnetizing factors obtained by solving Eq. 23 are given by:

$$\tilde{n}_{11}^{\dagger} = \frac{1}{m^2 - 1} \left\{ \frac{m}{(m^2 - 1)^{\frac{1}{2}}} \ln \left[ m + \left(m^2 - 1\right)^{\frac{1}{2}} \right] - 1 \right\} \tag{30}$$

and

$$\tilde{n}_{22}^{\dagger} = \frac{1}{2} \left( 1 - \tilde{n}_{11}^{\dagger} \right) , \tag{31}$$

where $\tilde{n}_{33}^{\dagger} = \tilde{n}_{22}^{\dagger}$, with $\tilde{n}_{11}^{\dagger}$ defined in Eq. 30 and $m = a/b$. The detailed deduction of the demagnetizing factors $\tilde{n}_{11}^{\dagger}$ (Eq. 30) and $\tilde{n}_{22}^{\dagger}$ (Eq. 31) can be found, for example, in Stoner (1945). These formulas were posteriorly presented by Emerson et al. (1985). It can be shown that these demagnetizing factors satisfy the conditions $\tilde{n}_{11}^{\dagger} + 2\, \tilde{n}_{22}^{\dagger} = 1$ and $\tilde{n}_{11}^{\dagger} < \tilde{n}_{22}^{\dagger}$.





**Oblate ellipsoids**

For oblate ellipsoids (e.g., $a < b = c$), the demagnetizing factors obtained by solving Eq. 23 are given by:

$$\tilde{n}_{11}^{\dagger} = \frac{1}{1-m^2} \left[ 1 - \frac{m}{(1-m^2)^{\frac{1}{2}}} \cos^{-1} m \right] \tag{32}$$

and

$$\tilde{n}_{22}^{\dagger} = \frac{1}{2} \left( 1 - \tilde{n}_{11}^{\dagger} \right) , \tag{33}$$

where $\tilde{n}_{33}^{\dagger} = \tilde{n}_{22}^{\dagger}$, with $\tilde{n}_{11}^{\dagger}$ defined in Eq. 32 and $m = a/b$. The detailed deduction of these demagnetizing factors can be found, for example, in Stoner (1945). These formulas can also be found in Emerson et al. (1985). The only difference, however, is that Emerson et al. (1985) replaced the term $\cos^{-1}$ by a term $\tan^{-1}$, according to the trigonometric identity $\tan^{-1} x = \cos^{-1}(1/\sqrt{x^2+1})$, $x > 0$. It can be shown that these demagnetizing factors satisfy the conditions $\tilde{n}_{11}^{\dagger} + 2\,\tilde{n}_{22}^{\dagger} = 1$ and $\tilde{n}_{11}^{\dagger} > \tilde{n}_{22}^{\dagger}$.

### 2.4.2 Depolarization tensor $\tilde{\mathbf{N}}^{\ddagger}(\tilde{\mathbf{r}})$

The elements $\tilde{n}_{ij}^{\ddagger}(\tilde{\mathbf{r}})$, $i = 1, 2, 3$, $j = 1, 2, 3$, of the transformed depolarization tensor $\tilde{\mathbf{N}}^{\ddagger}(\tilde{\mathbf{r}})$ are calculated according to Eq. 22, with $\tilde{f}(\tilde{\mathbf{r}})$ given by $\tilde{f}^{\ddagger}(\tilde{\mathbf{r}})$ (Eq. 20). By following Clark et al. (1986), the diagonal elements $\tilde{n}_{ii}^{\ddagger}(\tilde{\mathbf{r}})$ and the off-diagonal elements $\tilde{n}_{ij}^{\ddagger}(\tilde{\mathbf{r}})$, $i = 1, 2, 3$, $j = 1, 2, 3$, are given by

$$\tilde{n}_{ii}^{\ddagger}(\tilde{\mathbf{r}}) = -\frac{abc}{2} \left( \frac{\partial \lambda}{\partial \tilde{r}_i} h_i \tilde{r}_i + g_i \right) \tag{34}$$

and

$$\tilde{n}_{ij}^{\ddagger}(\tilde{\mathbf{r}}) = -\frac{abc}{2} \left( \frac{\partial \lambda}{\partial \tilde{r}_i} h_j \tilde{r}_j \right) , \tag{35}$$

where

$$h_i = -\frac{1}{(e_i^2 + \lambda)\,R(\lambda)} , \tag{36}$$

$$g_i = \int_{\lambda}^{\infty} \frac{1}{(e_i^2 + u)\,R(u)} du , \tag{37}$$

$e_1 = a$, $e_2 = b$, $e_3 = c$, and $\frac{\partial \lambda}{\partial \tilde{r}_i}$ is defined in Appendix B (Eq. B22). The functions $g_i$ (Eq. 37) are defined according to the ellipsoid type. Notice that the elements $\tilde{n}_{ii}^{\ddagger}(\tilde{\mathbf{r}})$ (Eq. 34) and $\tilde{n}_{ij}^{\ddagger}(\tilde{\mathbf{r}})$ (Eq. 35) are proportional to the second derivatives of the function $\tilde{f}^{\ddagger}(\tilde{\mathbf{r}})$ (Eq. 20), which is harmonic. Consequently, the diagonal elements $\tilde{n}_{ii}^{\ddagger}(\tilde{\mathbf{r}})$ satisfy the condition $\tilde{n}_{11}^{\ddagger}(\tilde{\mathbf{r}}) + \tilde{n}_{22}^{\ddagger}(\tilde{\mathbf{r}}) + \tilde{n}_{33}^{\ddagger}(\tilde{\mathbf{r}}) = 0$ for any point $\tilde{\mathbf{r}}$ outside the ellipsoid, independently of the ellipsoid type.





**Triaxial ellipsoids**

For triaxial ellipsoids (e.g., $a > b > c$), the functions $g_i$ (Eq. 37) are defined as follows:

$$g_1 = \frac{2}{(a^2 - b^2)(a^2 - c^2)^{\frac{1}{2}}} \left[ F(\kappa, \phi) - E(\kappa, \phi) \right],$$ (38)

$$g_2 = \frac{2(a^2 - c^2)^{\frac{1}{2}}}{(a^2 - b^2)(b^2 - c^2)} \left\{ E(\kappa, \phi) - \left( \frac{b^2 - c^2}{a^2 - c^2} \right) F(\kappa, \phi) - \frac{a^2 - b^2}{(a^2 - c^2)^{\frac{1}{2}}} \left[ \frac{c^2 + \lambda}{(a^2 + \lambda)(b^2 + \lambda)} \right]^{\frac{1}{2}} \right\}$$ (39)

and

$$g_3 = \frac{2}{(b^2 - c^2)(a^2 - c^2)^{\frac{1}{2}}} E(\kappa, \phi) + \frac{2}{b^2 - c^2} \left[ \frac{b^2 + \lambda}{(a^2 + \lambda)(c^2 + \lambda)} \right]^{\frac{1}{2}},$$ (40)

where $F(\kappa, \phi)$ and $E(\kappa, \phi)$ are defined by Eqs. 29 and 28, but with $\sin \phi = \sqrt{(a^2 - c^2)/(a^2 + \lambda)}$. A detailed deduction of these formulas was presented by Tejedor et al. (1995). Similar formulas can also be found in Clark et al. (1986).

**Prolate ellipsoids**

For prolate (e.g., $a > b = c$) ellipsoids, the functions $g_i$ (Eq. 37) are given by:

$$g_1 = \frac{2}{(a^2 - b^2)^{\frac{3}{2}}} \left\{ \ln \left[ \frac{(a^2 - b^2)^{\frac{1}{2}} + (a^2 + \lambda)^{\frac{1}{2}}}{(b^2 + \lambda)^{\frac{1}{2}}} \right] - \left( \frac{a^2 - b^2}{a^2 + \lambda} \right)^{\frac{1}{2}} \right\}$$ (41)

and

$$g_2 = \frac{1}{(a^2 - b^2)^{\frac{3}{2}}} \left\{ \frac{\left[ (a^2 - b^2)(a^2 + \lambda) \right]^{\frac{1}{2}}}{b^2 + \lambda} - \ln \left[ \frac{(a^2 - b^2)^{\frac{1}{2}} + (a^2 + \lambda)^{\frac{1}{2}}}{(b^2 + \lambda)^{\frac{1}{2}}} \right] \right\},$$ (42)

where $g_3 = g_2$. These formulas can be obtained by properly manipulating those presented by (Emerson et al., 1985).

**Oblate ellipsoids**

For oblate (e.g., $a < b = c$) ellipsoids, the functions $g_i$ (Eq. 37) are given by:

$$g_1 = \frac{2}{(b^2 - a^2)^{\frac{3}{2}}} \left\{ \left( \frac{b^2 - a^2}{a^2 + \lambda} \right)^{\frac{1}{2}} - \tan^{-1} \left[ \left( \frac{b^2 - a^2}{a^2 + \lambda} \right)^{\frac{1}{2}} \right] \right\}$$ (43)

and

$$g_2 = \frac{1}{(b^2 - a^2)^{\frac{3}{2}}} \left\{ \tan^{-1} \left[ \left( \frac{b^2 - a^2}{a^2 + \lambda} \right)^{\frac{1}{2}} \right] - \frac{\left[ (b^2 - a^2)(a^2 + \lambda) \right]^{\frac{1}{2}}}{b^2 + \lambda} \right\},$$ (44)

where $g_3 = g_2$. Similarly to the case of prolate ellipsoid shown previously, these formulas can be obtained by properly manipulating those presented by (Emerson et al., 1985).





### 2.5 Internal magnetic field and magnetization

By considering $\tilde{\mathbf{r}}$ as a point within the volume $\vartheta$ of the ellipsoid and using the Maxwell's postulate about the uniformity of the magnetic field $\mathbf{H}(\mathbf{r})$ inside ellipsoidal bodies, we can use Eq. 21 for defining the resultant uniform magnetic field $\tilde{\mathbf{H}}^{\dagger}$ inside the ellipsoidal body as follows:

$$\tilde{\mathbf{H}}^{\dagger} = \left(\mathbf{I} + \tilde{\mathbf{N}}^{\dagger}\tilde{\mathbf{K}}\right)^{-1}\tilde{\mathbf{H}}_0 \, , \tag{45}$$

where $\mathbf{I}$ is the identity matrix and $\tilde{\mathbf{N}}^{\dagger}$ is defined in the previous section.

Let us pre-multiply the uniform internal field $\tilde{\mathbf{H}}^{\dagger}$ (Eq. 45) by the transformed susceptibility tensor $\tilde{\mathbf{K}}$ (Eq. 21) to obtain

$$\begin{aligned}
\tilde{\mathbf{M}} &= \tilde{\mathbf{K}}\left(\mathbf{I} + \tilde{\mathbf{N}}^{\dagger}\tilde{\mathbf{K}}\right)^{-1}\tilde{\mathbf{H}}_0 \\
&= \left(\mathbf{I} + \tilde{\mathbf{K}}\tilde{\mathbf{N}}^{\dagger}\right)^{-1}\tilde{\mathbf{K}}\tilde{\mathbf{H}}_0 \, ,
\end{aligned} \tag{46}$$

where $\tilde{\mathbf{M}}$ represents the transformed magnetization, as can be easily verified by using Eqs. 12 and 21. The matrix identity used for obtaining the second line of Eq. 46 is given by Searle (1982, p. 151). Equation 46 can be easily generalized for the case in which the ellipsoid has also a uniform remanent magnetization $\tilde{\mathbf{M}}_R$. Let us first consider that the uniform remanent magnetization satisfies the condition $\tilde{\mathbf{H}}_A = \tilde{\mathbf{K}}^{-1}\tilde{\mathbf{M}}_R$, where $\tilde{\mathbf{H}}_A$ represents a hypothetical uniform ancient field. Then, if we assume that $\tilde{\mathbf{H}}_0$, in Eqs. 45 and 46, is in fact the sum of the inducing magnetic field $\tilde{\mathbf{H}}_0$ and the hypothetical ancient field $\tilde{\mathbf{H}}_A$, we obtain the following generalized equation

$$\mathbf{M} = \mathbf{\Lambda}\left(\mathbf{K}\mathbf{H}_0 + \mathbf{M}_R\right) \, , \tag{47}$$

where

$$\mathbf{\Lambda} = \mathbf{V}\left(\mathbf{I} + \tilde{\mathbf{K}}\tilde{\mathbf{N}}^{\dagger}\right)^{-1}\mathbf{V}^{\top} \, . \tag{48}$$

Despite of the coordinate system transformation represented by the matrix $\mathbf{V}$ (Eq. 2), Eq. 47 is consistent with that given by Clark et al. (1986, Eq. 38). It shows the combined effect of the anisotropy of magnetic susceptibility (AMS) and the shape anisotropy. The AMS is represented by the susceptibility tensor $\mathbf{K}$ (Eq. 13) and reflects the preferred orientation of the magnetic minerals forming the body. The susceptibility tensor appears in Eq. 47, defined in the main coordinate system (Fig. 1a), and Eq. 48, defined in the local coordinate system (Fig. 1b). The shape anisotropy is represented, in Eq. 47, by the depolarization tensor $\tilde{\mathbf{N}}^{\dagger}$ and reflects the self-demagnetization associated to the body shape. Notice that the resultant magnetization $\mathbf{M}$ (Eq. 47) does not necessarily have the same direction as the inducing field $\mathbf{H}_0$ (Eq. 9). The angular difference between the resultant magnetization and the inducing field depends on the combined effect of the anisotropy of magnetic susceptibility and the shape anisotropy.

For the particular case in which the susceptibility is isotropic, the susceptibility tensor is defined according to Eq. 14. In this case, the magnetization $\mathbf{M}$ (Eqs. 12 and 47), referred to the main coordinate system (Fig. 1a), and the matrix $\mathbf{\Lambda}$ (Eq. 48) can be rewritten as follows:

$$\mathbf{M} = \mathbf{\Lambda}\left(\chi\mathbf{H}_0 + \mathbf{M}_R\right) \, , \tag{49}$$

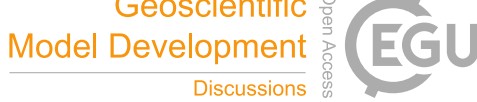



and

$$\mathbf{\Lambda} = \mathbf{V} \left( \mathbf{I} + \chi \tilde{\mathbf{N}}^{\dagger} \right)^{-1} \mathbf{V}^{\top} . \tag{50}$$

Despite the coordinate transformation represented by matrix $\mathbf{V}$ (Eq. 2), this equation is in perfect agreement with those presented by Guo et al. (2001, Eqs. 13–15). The first term, depending on the inducing field $\mathbf{H}_0$ (Eq. 9), represents the induced

magnetization whereas the term depending on $\mathbf{M}_R$ is the remanent magnetization. Equation 49 reveals that, as pointed out by many authors (e.g., Maxwell, 1873; DuBois, 1896; Stoner, 1945; Clark et al., 1986; Stratton, 2007), the induced magnetization opposes the inducing field if it is parallel to an ellipsoid axis, independently of the ellipsoid type. Otherwise, the magnetization is not necessarily parallel to the inducing field. If we additionally consider that $\chi << 1$, the matrix $\mathbf{\Lambda}$ (Eq. 50) approaches to the identity and the magnetization $\mathbf{M}$ (Eq. 49) can be approximated by:

$$\breve{\mathbf{M}} = \chi \mathbf{H}_0 + \mathbf{M}_R , \tag{51}$$

which is the classical equation describing the resultant magnetization in applied geophysics (Blakely, 1996, p. 89). Notice that, in this particular case, the induced magnetization is parallel to the inducing field $\mathbf{H}_0$ (Eq. 9), whether it is parallel to an ellipsoid axis or not. Usually, Eq. 51 is considered a good approximation for $\chi \leq 0.1$ SI. Although this value has been largely used in the literature, there have been few empirical and/or theoretical investigations about it.

### 2.5.1  Relationship between $\chi$ and the relative error in $\breve{\mathbf{M}}$

In the case of isotropic susceptibility, the resultant magnetization $\mathbf{M}$ (Eq. 49) may be determined by solving the following linear system:

$$\mathbf{\Lambda}^{-1} \mathbf{M} = \chi \mathbf{H}_0 + \mathbf{M}_R , \tag{52}$$

where, according to Eq. 50,

$$\mathbf{\Lambda}^{-1} = \mathbf{V} \left( \mathbf{I} + \chi \tilde{\mathbf{N}}^{\dagger} \right) \mathbf{V}^{\top} . \tag{53}$$

As we have already pointed out, the approximated magnetization $\breve{\mathbf{M}}$ (Eq. 51) represents the particular case in which the matrix $\mathbf{\Lambda}$ (Eq. 50), and consequently the matrix $\mathbf{\Lambda}^{-1}$ (Eq. 53), are close to the identity.

   Consider a perturbed matrix $\delta \mathbf{\Lambda}^{-1}$ given by

$$\delta \mathbf{\Lambda}^{-1} = \mathbf{\Lambda}^{-1} - \mathbf{I} \tag{54}$$

and, similarly, a perturbed magnetization vector $\delta \mathbf{M}$ given by

$$\delta \mathbf{M} = \mathbf{M} - \breve{\mathbf{M}} . \tag{55}$$

By using these two equations, we may rewrite that of the approximated magnetization $\breve{\mathbf{M}}$ (Eq. 51) as follows:

$$\left( \mathbf{\Lambda}^{-1} - \delta \mathbf{\Lambda}^{-1} \right) \left( \mathbf{M} - \delta \mathbf{M} \right) = \chi \mathbf{H}_0 + \mathbf{M}_R . \tag{56}$$





Now, by subtracting the true magnetization $\mathbf{M}$ (Eq. 52) from this linear system (Eq. 56) and rearranging the terms, we obtain the following linear system for the perturbed magnetization $\delta\mathbf{M}$ (Eq. 55):

$$\delta\mathbf{M} = -\delta\mathbf{\Lambda}^{-1}\mathbf{M}\,. \tag{57}$$

By using the concept of vector norm and its corresponding operator norm (Demmel, 1997; Golub and Loan, 2013), we may use Eq. 57 to write the following inequality:

$$\frac{\|\delta\mathbf{M}\|}{\|\mathbf{M}\|} \leq \|\delta\mathbf{\Lambda}^{-1}\|\,. \tag{58}$$

where $\|\delta\mathbf{M}\|$ and $\|\mathbf{M}\|$ denote Euclidean norms (or 2-norms) and the term $\|\delta\mathbf{\Lambda}^{-1}\|$ denotes the matrix 2-norm of $\delta\mathbf{\Lambda}^{-1}$. By using Eqs. 53 and 54 and the orthogonal invariance of the matrix 2-norm (Demmel, 1997; Golub and Loan, 2013), we define $\|\delta\mathbf{\Lambda}^{-1}\|$ as follows:

$$\|\delta\mathbf{\Lambda}^{-1}\| = \chi\tilde{n}_{max}^{\dagger}\,, \tag{59}$$

where $\tilde{n}_{max}^{\dagger}$ is the demagnetization factor associated to the shortest ellipsoid semi-axis. For a triaxial ellipsoid, $\tilde{n}_{max}^{\dagger} \equiv \tilde{n}_{33}^{\dagger}$ (Eq. 27), for a prolate ellipsoid, $\tilde{n}_{max}^{\dagger} \equiv \tilde{n}_{22}^{\dagger}$ (Eq. 31), and, for an oblate ellipsoid, $\tilde{n}_{max}^{\dagger} \equiv \tilde{n}_{11}^{\dagger}$ (Eq. 32). It is worth stressing that, independently of the ellipsoid type, $\tilde{n}_{max}^{\dagger}$ is a scalar function of the ellipsoid semi-axes. In Eq. 58, the ratio $\|\delta\mathbf{M}\|\,\|\mathbf{M}\|^{-1}$ represents the *relative error* in the approximated magnetization $\breve{\mathbf{M}}$ (Eq. 51) with respect to the true magnetization $\mathbf{M}$ (Eqs. 49 and 52). Given a target relative error $\epsilon$ and an ellipsoid with given semi-axes, we may use the inequality represented by Eq. 59 to define

$$\chi_{max} = \frac{\epsilon}{\tilde{n}_{max}^{\dagger}}\,, \tag{60}$$

which represents the maximum isotropic susceptibility that the ellipsoidal body can assume in order o guarantee a relative error lower than or equal to $\epsilon$. For isotropic susceptibilities greater than $\chi_{max}$, there is no guarantee that the relative error in the approximated magnetization $\breve{\mathbf{M}}$ (Eq. 51) with respect to the true magnetization $\mathbf{M}$ (Eqs. 49 and 52) is lower than or equal to $\epsilon$. The geoscientific community has been using $\chi_{max} = 0.1$ SI as a limit value for neglecting the self-demagnetization and, consequently, use magnetization $\breve{\mathbf{M}}$ (Eq. 51) as a good approximation of the true magnetization $\mathbf{M}$ (Eqs. 49 and 52). Equation 60, on the other hand, defines $\chi_{max}$ as a function of the ellipsoid semi-axes, according to a user-specified relative error $\epsilon$.

## 2.6 External magnetic field and total-field anomaly

The magnetic field $\Delta\mathbf{H}(\mathbf{r})$ produced by an ellipsoid at external points is calculated from Eqs. 21 and 47 as the difference between the resultant field $\mathbf{H}(\mathbf{r})$ and the inducing field $\mathbf{H}_0$:

$$\Delta\mathbf{H}(\mathbf{r}) = \mathbf{V}\tilde{\mathbf{N}}^{\ddagger}(\tilde{\mathbf{r}})\,\mathbf{V}^{\top}\mathbf{M}\,, \tag{61}$$

where $\tilde{\mathbf{N}}^{\ddagger}(\tilde{\mathbf{r}})$ is the transformed depolarization tensor whose elements $\tilde{n}_{ii}^{\ddagger}(\tilde{\mathbf{r}})$ and $\tilde{n}_{ij}^{\ddagger}(\tilde{\mathbf{r}})$ are defined, respectively, by Eqs. 34 and 35. $\Delta\mathbf{H}(\mathbf{r})$ represents the magnetic field produced by a uniformly magnetized body located in the crust. Equation 61 gives




the magnetic field (in $\mathrm{A\,m^{-1}}$) produced by an ellipsoid. However, in geophysics, the most widely used field is the magnetic induction (in nT). Fortunately, this conversion can be easily done by multiplying Eq. 61 by $k_m = 10^9\,\mu_0$, where $\mu_0$ represents the magnetic constant (in $\mathrm{H\,m^{-1}}$). For geophysical applications, it is preferable to calculate the total-field anomaly produced by the magnetic sources. This scalar quantity is given by (Blakely, 1996):

$$\Delta T(\mathbf{r}) = \|\mathbf{B}_0 + \Delta \mathbf{B}(\mathbf{r})\| - \|\mathbf{B}_0\|\,, \tag{62}$$

where $\mathbf{B}_0 = k_m\,\mathbf{H}_0$ and $\Delta\mathbf{B}(\mathbf{r}) = k_m\,\Delta\mathbf{H}(\mathbf{r})$, with $\mathbf{H}_0$ and $\Delta\mathbf{H}(\mathbf{r})$ defined, respectively, by Eqs. 9 and 61. In practical situations, however, $\|\mathbf{B}_0\| >> \|\Delta\mathbf{B}(\mathbf{r})\|$ and, consequently, the following approximation is valid (Blakely, 1996):

$$\Delta T(\mathbf{r}) \approx \frac{\mathbf{B}_0^\top \Delta \mathbf{B}(\mathbf{r})}{\|\mathbf{B}_0\|}\,. \tag{63}$$

## 3   Computational implementation and reproducibility

The code is implemented in the Python language, by using the NumPy and SciPy libraries (van der Walt et al., 2011), as part of the open-source source library Fatiando a Terra (Uieda et al., 2013). Our code is very modular and has a test suite formed by a considerable number of assertions, unit tests, doc tests, and integration tests. We refer the readers interested in best practices for scientific computing to Wilson et al. (2014).

The numerical simulations presented here were generated with the Jupyter Notebook (http://jupyter.org), which is a web application that allows creating and sharing documents that contain live code, equations, visualizations and explanatory text. Besides using Fatiando a Terra (Uieda et al., 2013), the numerical simulations use the NumPy library (van der Walt et al., 2011) to perform numerical computations and the Matplotlib library (Hunter, 2007) to plot the results and generate figures. The Jupyter Notebooks used to produce all the results presented here are available in a repository on GitHub (https://github.com/pinga-lab/magnetic-ellipsoid).

## 4   Numerical simulations

All the code developed for generating the results presented in the following subsections, as well as the code developed for generating additional numerical simulations, can be found at the folder *code* of the online repository https://github.com/pinga-lab/magnetic-ellipsoid.

### 4.1   Demagnetizing factors

We simulate a triaxial ellipsoid with semi-axes $a_0 = 1000$ m, $b_0 = 700$ m, and $c_0 = 200$ m. Then we use this ellipsoid as a reference to generate 100 different triaxial ellipsoids and calculate their demagnetizing factors $\tilde{n}_{11}^\dagger$, $\tilde{n}_{22}^\dagger$, and $\tilde{n}_{33}^\dagger$ by using Eqs. 25, 26, and 27. The semi-axes of these 100 ellipsoids are given by $a = a_0 + u\,b_0$, $b = b_0 + u\,b_0$, and $c = c_0 + u\,b_0$, where $0 \le u \le 10$. Notice that, for $u = 0$, the resulting semi-axes are equal those of the reference ellipsoid. The larger the variable $u$, the larger the resulting semi-axes $a$, $b$, and $c$, but the smaller the relative difference between them. Consequently, the resulting





ellipsoids obtained from the semi-axes $a$, $b$, and $c$ become more spherical as $u$ increases. In this case, the demagnetizing factors $\tilde{n}_{11}^{\dagger}$ (Eq. 25), $\tilde{n}_{22}^{\dagger}$ (Eq. 26), and $\tilde{n}_{33}^{\dagger}$ (Eq. 27) tend to $1/3$ (e.g., Stoner, 1945).

Figure 2a shows the calculated demagnetizing factors $\tilde{n}_{11}^{\dagger}$ (in red), $\tilde{n}_{22}^{\dagger}$ (in green), and $\tilde{n}_{33}^{\dagger}$ (in blue) for the 100 triaxial ellipsoids. The result shows that the relative difference between the demagnetizing factors is large for small values of $u$ and

decreases as $u$ increases. In this case, all demagnetizing factors tend to $1/3$, according to what we know from theory. Besides, Fig. 2a confirms that the demagnetizing factors satisfy the condition $\tilde{n}_{11}^{\dagger} < \tilde{n}_{22}^{\dagger} < \tilde{n}_{33}^{\dagger}$ independently of the value of $u$.

We have also simulated 100 different prolate ellipsoids with semi-axes $a = m\, b_0$ and $b = b_0$, where $1.02 \le m \le 10$ and $b_0 = 1000$ m, and calculate their demagnetizing factors $\tilde{n}_{11}^{\dagger}$ and $\tilde{n}_{22}^{\dagger}$ by using Eqs. 30 and 31, respectively. Similarly, we simulated 100 different oblate ellipsoids with semi-axes $a = m\, b_0$ and $b = b_0$, where $0.02 \le m \le 0.98$ and $b_0 = 1000$ m, and

calculate their demagnetizing factors $\tilde{n}_{11}^{\dagger}$ and $\tilde{n}_{22}^{\dagger}$ by using Eqs. 32 and 33, respectively.

Figures 2b and c show the results obtained for the 100 prolate and the 100 oblate ellipsoids, respectively. As expected from theory, the demagnetizing factors $\tilde{n}_{11}^{\dagger}$ (red line in Fig. 2b) and $\tilde{n}_{22}^{\dagger}$ (green line in Fig. 2b) calculated for the prolate ellipsoids are close to $1/3$ for $m$ close to 1. Besides, these demagnetizing factors satisfy the condition $\tilde{n}_{11}^{\dagger} < \tilde{n}_{22}^{\dagger}$ for all values of $m$. The result obtained for the oblate ellipsoids (Fig. 2c) are also in perfect agreement with theory. The demagnetizing factors $\tilde{n}_{11}^{\dagger}$ (in

red) and $\tilde{n}_{22}^{\dagger}$ (in green), which were calculated by using Eqs. 32 and 33, respectively, are close to $1/3$ for $m$ close to 0 and satisfy the condition $\tilde{n}_{11}^{\dagger} > \tilde{n}_{22}^{\dagger}$ for all values of $m$.

### 4.2 Simulation of a geological body

We simulated an ellipsoidal body similar to the Warrego orebody, which was the resource on which the well-known Warrego mine was developed in Tennant Creek, Australia. After nearly a decade as one of the most important gold and copper mines in

Australia, the Warrego mine was closed in late 1989. According to Wedekind (1990), the Warrego orebody is a combination of two major and several small ironstone lodes, which are discrete bodies comprised predominantly of magnetite or hematite above the base of oxidation. Farrar (1979) represented the Warrego orebody as a triaxial ellipsoid having a high isotropic susceptibility. In this case, the self-demagnetization strongly impacts the magnetic modelling of this body.

Table 1 shows the parameters defining a synthetic orebody which is based on that presented by Farrar (1979) to represent

the Warrego orebody. Figure 3 shows the total-field anomaly $\Delta T(\mathbf{r})$ (Eq. 63) produced by the synthetic body on a regular grid of $100 \times 100$ points at a constant vertical coordinate $z = 0$ m. The total-field anomaly varies from $\approx -71$ nT to $\approx 482$ nT, resulting in a peak-to-peak amplitude of $\approx 553$ nT, and was calculated by using the true magnetization $\mathbf{M}$ defined in Eqs. 49 and 52.

We have calculated the difference between the total-field anomaly $\Delta T(\mathbf{r})$ (Eq. 63) calculated with the true magnetization

$\mathbf{M}$ (Eqs. 49 and 52) and that calculated with the approximated magnetization $\breve{\mathbf{M}}$ (Eq. 51). The differences were calculated by using the synthetic body defined in Tab. 1, but with three different isotropic susceptibilities. Figures 4a, 4b and 4c show the differences calculated by using, respectively, isotropic susceptibilities $\chi = 1.69$ SI (Tab. 1), $\chi_1 = 0.1$ SI and $\chi_2 = 0.116$ SI.

As expected, the differences calculated by using the higher isotropic susceptibility (Fig. 4a) are very large. The peak-to-peak amplitude is $\approx 40$ nT and represents $\approx 8\%$ of the peak-to-peak amplitude of the total-field anomaly shown in Fig. 3.



Figure 4b shows the differences calculated by using $\chi_1 = 0.1$ SI. It is commonly accepted that, for bodies having isotropic susceptibilities lower than or equal to $0.1$ SI, the self-demagnetization can be neglected and, consequently, the magnetization $\breve{\mathbf{M}}$ (Eq. 51) is a good approximation of the true magnetization $\mathbf{M}$ (Eqs. 49 and 52). In our test, the use of $\chi_1 = 0.1$ SI leads to a relative error $\|\delta\mathbf{M}\| \, \|\mathbf{M}\|^{-1} \approx 0.7\%$ (Eq. 58) in the magnetization. The peak-to-peak amplitude of the differences in the total-field anomaly (Fig. 4b) is $\approx 0.2$ nT, which represents $\approx 0.6\%$ of the peak-to-peak amplitude of the total-field anomaly calculates by using the true magnetization $\mathbf{M}$ (Eqs. 49 and 52).

Finally, Fig. 4c shows the differences calculated by using $\chi_2 = 0.116$ SI. This value was calculated by using Eq. 60 with a target relative error $\epsilon = 8\%$ and the $\tilde{n}^\dagger_{max}$ defined by Eq. 27. By using this isotropic susceptibility, it is expected that the calculated relative error $\|\delta\mathbf{M}\| \, \|\mathbf{M}\|^{-1}$ (Eq. 58) in the magnetization be lower than or equal to the target relative error $\epsilon = 8\%$. In this test, the use of $\chi_2 = 0.116$ SI leads to a relative error $\|\delta\mathbf{M}\| \, \|\mathbf{M}\|^{-1} \approx 0.8\%$ (Eq. 58) in the magnetization. The peak-to-peak amplitude of the differences in the total-field anomaly (Fig. 4c) is $\approx 0.3$ nT, which represents $\approx 0.7\%$ of the peak-to-peak amplitude of the total-field anomaly calculates by using the true magnetization $\mathbf{M}$ (Eqs. 49 and 52). In this case, the use an isotropic susceptibility greater than the usual limit $0.1$ SI does not mislead the magnetic modelling dramatically. On the contrary, it shows small discrepancies in the magnetic modelling and validates Eq. 60.

## 5 Conclusions

We present an integrated review of the vast literature about the magnetic modelling of triaxial, prolate and oblate ellipsoids. We also present a theoretical discussion about the determination of the isotropic susceptibility value above which the self-demagnetization must be taken into consideration. We propose an alternative way of determining this value based on the body shape and the maximum relative error allowed in the resultant magnetization. Our approach is an alternative to the constant value which seems to be determined empirically and has been used by the geoscientific community. Our alternative approach is validated by the results obtained with numerical simulations. In a future work, it would be interesting to use a similar approach to to bound the maximum relative error in the magnetic field calculated by neglecting the self-demagnetization.

This work also provide a set of routines to model the magnetic field produced by ellipsoids. The routines are written in Python language as part of the Fatiando a Terra (Uieda et al., 2013), which is an open-source library for modelling and inversion in geophysics. The current version of our code is freely distributed in a repository of GitHub. We are working to integrate our routines in the following stable release of Fatiando a Terra. We hope that these routines be useful to the wide geoscientific community either in researching and teaching.

## 6 Code availability

The current version of our code is freely distributed, under the BSD 3-clause licence, in a repository on GitHub (https://github. com/pinga-lab/magnetic-ellipsoid). Instructions for running the current version of our code are also provided in the repository. The code was developed as part of the Fatiando a Terra (Uieda et al., 2013) open-source Python library for modelling and



inversion in geophysics. Documentation and installation instructions for the 0.5 release version of Fatiando a Terra are provided at http://www.fatiando.org/v0.5.

## Appendix A: Derivatives of the functions $f(\mathbf{r})$ and $\tilde{f}(\tilde{\mathbf{r}})$

Let $\tilde{f}(\tilde{\mathbf{r}})$ be the scalar function obtained by transforming $f(\mathbf{r})$ (Eq. 17) from the main coordinate system (Fig. 1a) to the local

coordinate system (Fig. 1b). For convenience, let us rewrite Eq. 8 as follows:

$$\tilde{r}_k = v_{k1} r_1 + v_{k2} r_2 + v_{k3} r_3 + c_k \,, \tag{A1}$$

where $\tilde{r}_k$, $k = 1, 2, 3$, are the elements of the transformed position vector $\tilde{\mathbf{r}}$ (Eq. 8), $r_j$, $j = 1, 2, 3$, are the elements of the position vector $\mathbf{r}$ (Eq. 1), $v_{kj}$, $j = 1, 2, 3$, are the elements of the matrix $\mathbf{V}$ (Eq. 2), and $c_k$ is a constant defined by the coordinates $x_c$, $y_c$, and $z_c$ of the centre of the ellipsoidal body.

By considering the functions $f(\mathbf{r})$ (Eq. 17) and $\tilde{f}(\tilde{\mathbf{r}})$ evaluated at the same point, but on different coordinate systems, we have:

$$\frac{\partial f(\mathbf{r})}{\partial r_j} = \frac{\partial \tilde{f}(\tilde{\mathbf{r}})}{\partial \tilde{r}_1} \frac{\partial \tilde{r}_1}{\partial r_j} + \frac{\partial \tilde{f}(\tilde{\mathbf{r}})}{\partial \tilde{r}_2} \frac{\partial \tilde{r}_2}{\partial r_j} + \frac{\partial \tilde{f}(\tilde{\mathbf{r}})}{\partial \tilde{r}_3} \frac{\partial \tilde{r}_3}{\partial r_j} \,, \quad j = 1, 2, 3 \,,$$

which, from Eq. A1, can be given by

$$\frac{\partial f(\mathbf{r})}{\partial r_j} = v_{j1} \frac{\partial \tilde{f}(\tilde{\mathbf{r}})}{\partial \tilde{r}_1} + v_{j2} \frac{\partial \tilde{f}(\tilde{\mathbf{r}})}{\partial \tilde{r}_2} + v_{j3} \frac{\partial \tilde{f}(\tilde{\mathbf{r}})}{\partial \tilde{r}_3} \,, \quad j = 1, 2, 3 \,. \tag{A2}$$

Now, by deriving $\frac{\partial f(\mathbf{r})}{\partial r_j}$ (Eq. A2) with respect to the $i$th element $r_i$ of the position vector $\mathbf{r}$ (Eq. 1), we obtain:

$$\begin{aligned}
\frac{\partial^2 f(\mathbf{r})}{\partial r_i \partial r_j} &= v_{j1} \frac{\partial}{\partial r_i}\left( \frac{\partial \tilde{f}(\tilde{\mathbf{r}})}{\partial \tilde{r}_1} \right) + v_{j2} \frac{\partial}{\partial r_i}\left( \frac{\partial \tilde{f}(\tilde{\mathbf{r}})}{\partial \tilde{r}_2} \right) + v_{j3} \frac{\partial}{\partial r_i}\left( \frac{\partial \tilde{f}(\tilde{\mathbf{r}})}{\partial \tilde{r}_3} \right) \\
&= v_{j1} \left( \frac{\partial^2 \tilde{f}(\tilde{\mathbf{r}})}{\partial \tilde{r}_1 \partial \tilde{r}_1} v_{i1} + \frac{\partial^2 \tilde{f}(\tilde{\mathbf{r}})}{\partial \tilde{r}_2 \partial \tilde{r}_1} v_{i2} + \frac{\partial^2 \tilde{f}(\tilde{\mathbf{r}})}{\partial \tilde{r}_3 \partial \tilde{r}_1} v_{i3} \right) + \\
&\quad + v_{j2} \left( \frac{\partial^2 \tilde{f}(\tilde{\mathbf{r}})}{\partial \tilde{r}_1 \partial \tilde{r}_2} v_{i1} + \frac{\partial^2 \tilde{f}(\tilde{\mathbf{r}})}{\partial \tilde{r}_2 \partial \tilde{r}_2} v_{i2} + \frac{\partial^2 \tilde{f}(\tilde{\mathbf{r}})}{\partial \tilde{r}_3 \partial \tilde{r}_2} v_{i3} \right) + \\
&\quad + v_{j3} \left( \frac{\partial^2 \tilde{f}(\tilde{\mathbf{r}})}{\partial \tilde{r}_1 \partial \tilde{r}_3} v_{i1} + \frac{\partial^2 \tilde{f}(\tilde{\mathbf{r}})}{\partial \tilde{r}_2 \partial \tilde{r}_3} v_{i2} + \frac{\partial^2 \tilde{f}(\tilde{\mathbf{r}})}{\partial \tilde{r}_3 \partial \tilde{r}_3} v_{i3} \right) \\
&= \begin{bmatrix} v_{j1} & v_{j2} & v_{j3} \end{bmatrix} \tilde{\mathbf{F}}(\tilde{\mathbf{r}}) \begin{bmatrix} v_{i1} \\ v_{i2} \\ v_{i3} \end{bmatrix} \,,
\end{aligned} \tag{A3}$$

where $\tilde{\mathbf{F}}(\tilde{\mathbf{r}})$ is a $3 \times 3$ matrix whose $ij$-th element is $\frac{\partial^2 \tilde{f}(\tilde{\mathbf{r}})}{\partial \tilde{r}_i \partial \tilde{r}_j}$. From Eq. A3, we obtain

$$\mathbf{F}(\mathbf{r}) = \mathbf{V}\,\tilde{\mathbf{F}}(\tilde{\mathbf{r}})\,\mathbf{V}^\top \,, \tag{A4}$$





where $\mathbf{F}(\mathbf{r})$ is a $3 \times 3$ matrix whose $ij$-th element is $\frac{\partial^2 f(\mathbf{r})}{\partial r_i \partial r_j}$ and $\mathbf{V}$ (Eq. 2) is defined according to the ellipsoid type. As one may noticed, the matrices $\mathbf{F}(\mathbf{r})$ and $\tilde{\mathbf{F}}(\tilde{\mathbf{r}})$ represent the Hessians of the functions $f(\mathbf{r})$ (Eq. 17) and $\tilde{f}(\tilde{\mathbf{r}})$, respectively. Besides, the depolarization tensor $\mathbf{N}(\mathbf{r})$ (Eq. 15) can be rewritten by using the matrix $\mathbf{F}(\mathbf{r})$ as follows

$$\mathbf{N}(\mathbf{r}) = \frac{1}{4\pi} \mathbf{F}(\mathbf{r}) \,. \tag{A5}$$

By properly using the orthogonality of the matrix $\mathbf{V}$ (Eq. 2), we may rewrite Eq. A4 as follows:

$$\tilde{\mathbf{F}}(\tilde{\mathbf{r}}) = \mathbf{V}^\top \mathbf{F}(\mathbf{r}) \mathbf{V} \,. \tag{A6}$$

Finally, by multiplying both sides of Eq. A6 by $\frac{1}{4\pi}$ and using Eq. A5, we conclude that

$$\tilde{\mathbf{N}}(\tilde{\mathbf{r}}) = \mathbf{V}^\top \mathbf{N}(\mathbf{r}) \mathbf{V} \,. \tag{A7}$$

### Appendix B: Parameter $\lambda$ and its spatial derivatives

Here, we follow the reasoning presented by Webster (1904) for analysing the parameter $\lambda$ which defines triaxial, prolate and oblate ellipsoids.

### B1    Parameter $\lambda$ defining triaxial ellipsoids

Let us consider an ellipsoid with semi-axes $a$, $b$, $c$ oriented along the $\tilde{x}$-, $\tilde{y}$-, and $\tilde{z}$-axis, respectively, of its local coordinate system (Fig. 1b), where $a > b > c > 0$. This ellipsoid is defined by the following equation:

$$\frac{\tilde{x}^2}{a^2} + \frac{\tilde{y}^2}{b^2} + \frac{\tilde{z}^2}{c^2} = 1 \,. \tag{B1}$$

A quadric surface (e.g., ellipsoid, hyperboloid of one sheet or hyperboloid of two sheets) which is confocal with the ellipsoid defined in Eq. B1 can be described as follows:

$$\frac{\tilde{x}^2}{a^2 + u} + \frac{\tilde{y}^2}{b^2 + u} + \frac{\tilde{z}^2}{c^2 + u} = 1 \,, \tag{B2}$$

where $u$ is a real number. Equation B2 represents an ellipsoid for $u$ satisfying the condition

$$u + c^2 > 0 \,. \tag{B3}$$

Given $a$, $b$, $c$, and a $u$ satisfying B3, we may use B2 for determining a set of points $(x, y, z)$ lying on the surface of an ellipsoid which is confocal with that one defined in Eq. B1. Now, consider the problem of determining the ellipsoid which is confocal with that one defined in B1 and pass through a particular point $(\tilde{x}, \tilde{y}, \tilde{z})$. This problem consists in determining the real number $u$ that, given $a$, $b$, $c$, $\tilde{x}$, $\tilde{y}$, and $\tilde{z}$, satisfies Eq. B2 and the condition expressed by Eq. B3. By rearranging Eq. B2, we obtain the following cubic equation for $u$:

$$p(u) = (a^2 + u)(b^2 + u)(c^2 + u) - (b^2 + u)(c^2 + u)\tilde{x}^2 - (a^2 + u)(c^2 + u)\tilde{y}^2 - (a^2 + u)(b^2 + u)\tilde{z}^2 \,. \tag{B4}$$





This cubic equation shows that:

$$u = \begin{cases} d \to \infty & , \quad p(u) > 0 \\ -c^2 & , \quad p(u) < 0 \\ -b^2 & , \quad p(u) > 0 \\ -a^2 & , \quad p(u) < 0 \end{cases} . \tag{B5}$$

Notice that, according to B5, the smaller, intermediate and largest roots of the cubic equation $p(u)$ (Eq. B4) are located, respectively, in the intervals $[-a^2, -b^2]$, $[-b^2, -c^2]$ and $[-c^2, \infty[$. Remember that we are interested in a $u$ satisfying the

condition expressed by Eq. B3. Consequently, according to the signal analysis shown in Eq. B5, we are interested in the largest root $\lambda$ of the cubic equation $p(u)$ (Eq. B4).

From Eq. B4, we obtain a simpler one given by

$$p(u) = u^3 + p_2 \, u^2 + p_1 \, u + p_0 \,, \tag{B6}$$

where

$$p_2 = a^2 + b^2 + c^2 - \tilde{x}^2 - \tilde{y}^2 - \tilde{z}^2 \,, \tag{B7}$$

$$p_1 = b^2 \, c^2 + a^2 \, c^2 + a^2 \, b^2 - (b^2 + c^2) \, \tilde{x}^2 - (a^2 + c^2) \, \tilde{y}^2 - (a^2 + b^2) \, \tilde{z}^2 \tag{B8}$$

and

$$p_0 = a^2 \, b^2 \, c^2 - b^2 \, c^2 \, \tilde{x}^2 - a^2 \, c^2 \, \tilde{y}^2 - a^2 \, b^2 \, \tilde{z}^2 \,. \tag{B9}$$

Finally, from Eqs. B7, B8 and B9, the largest root $\lambda$ of $p(u)$ (Eq. B6) can be calculated as follows (Weisstein, 2017):

$$\lambda = 2 \, \sqrt{-Q} \, \cos\left(\frac{\varphi}{3}\right) - \frac{p_2}{3} \,, \tag{B10}$$

where

$$\varphi = \cos^{-1}\left(\frac{R}{\sqrt{-Q^3}}\right) \,, \tag{B11}$$

$$Q = \frac{3 \, p_1 - p_2^2}{9} \tag{B12}$$

and

$$R = \frac{9 \, p_1 \, p_2 - 27 \, p_0 - 2 \, p_2^3}{54} \,. \tag{B13}$$





## B2    Parameter $\lambda$ defining prolate and oblate ellipsoids

Let us now consider a prolate ellipsoid with semi-axes $a$, $b$, $c$ oriented along the $\tilde{x}$-, $\tilde{y}$-, and $\tilde{z}$-axis, respectively, of its local coordinate system (Fig. 1b), where $a > b = c > 0$. In this case, the Eq. defining the surface of the ellipsoid is obtained by substituting $c = b$ in Eq. B1. Consequently, the equation defining the respective confocal quadric surface is given by

$$\frac{\tilde{x}^2}{a^2 + u} + \frac{\tilde{y}^2 + \tilde{z}^2}{b^2 + u} = 1 \tag{B14}$$

and the new condition that must be fulfilled by the variable $u$ so that Eq. B14 represent an ellipsoid is

$$u + b^2 > 0 \,. \tag{B15}$$

Similarly to the case of a triaxial ellipsoid presented in the previous subsection, we are interested in determining the real number $u$ that, given $a$, $b$, $\tilde{x}$, $\tilde{y}$, and $\tilde{z}$, satisfies Eq. B14 and the condition expressed by Eq. B15. From Eq. B14, we obtain the following quadratic equation for $u$:

$$p(u) = (a^2 + u)(b^2 + u) - (b^2 + u)\,\tilde{x}^2 - (a^2 + u)\,(\tilde{y}^2 + \tilde{z}^2) \,. \tag{B16}$$

This equation shows that

$$u = \begin{cases} d \to \infty & , \quad f(\rho) > 0 \\ -b^2 & , \quad f(\rho) < 0 \\ -a^2 & , \quad f(\rho) > 0 \end{cases} \tag{B17}$$

and, consequently, that its two roots lie in the intervals $[-a^2, -b^2]$ and $[-b^2, \infty[$. Therefore, according to the condition established by Eq. B15 and the signal analysis shown in Eq. B17, we are interested in the largest root $\lambda$ of the quadratic equation $p(u)$ (Eq. B16).

By properly manipulating Eq. B16, we obtain a simpler one given by

$$p(u) = u^2 + p_1\,u + p_0 \,, \tag{B18}$$

where

$$p_1 = a^2 + b^2 - \tilde{x}^2 - \tilde{y}^2 - \tilde{z}^2 \tag{B19}$$

and

$$p_0 = a^2\,b^2 - b^2\,\tilde{x}^2 - a^2\,(\tilde{y}^2 + \tilde{z}^2) \,. \tag{B20}$$

Finally, by using Eqs. B19 and B20, the largest root $\lambda$ of $p(u)$ (Eq. B18) can be easily calculated as follows:

$$\lambda = \frac{-p_1 + \sqrt{p_1^2 - 4 p_0}}{2} \,. \tag{B21}$$





In the case of oblate ellipsoids, the procedure for determining the parameter $\lambda$ is very similar to this one for prolate ellipsoids. The semi-axes $a$, $b$, $c$ of oblate ellipsoids are defined so that $b = c > a > 0$ and the condition that must be fulfilled by the variable $u$ is $u + a^2 > 0$. In this case, the two roots of the resulting quadratic equation lie in the intervals $[-b^2, -a^2]$ and $[-a^2, \infty[$. Consequently, we are still interested in the largest root of the quadratic equation for the variable $u$, which is also calculated by using Eq. B21.

### B3   Spatial derivative of the parameter $\lambda$

The magnetic modelling of triaxial, prolate or oblate ellipsoids requires not only the parameter lambda defined by Eqs. B10 and B21, but also its derivatives with respect to the spatial coordinates $\tilde{x}$, $\tilde{y}$, and $\tilde{z}$. Fortunately, the spatial derivatives of the parameter $\lambda$ can be calculated in a very similar way for all ellipsoid types.

Let us first consider a triaxial ellipsoid. In this case, the spatial derivatives of $\lambda$ are given by

$$\frac{\partial \lambda}{\partial \tilde{r}_j} = \frac{\frac{2\tilde{r}_j}{(e_j^2 + \lambda)}}{\left(\frac{\tilde{x}}{a^2 + \lambda}\right)^2 + \left(\frac{\tilde{y}}{b^2 + \lambda}\right)^2 + \left(\frac{\tilde{z}}{c^2 + \lambda}\right)^2}, \quad j = 1, 2, 3, \tag{B22}$$

where $\tilde{r}_1 = \tilde{x}$, $\tilde{r}_2 = \tilde{y}$, $\tilde{r}_3 = \tilde{z}$, $e_1 = a$, $e_2 = b$, and $e_3 = c$. This equation can be determined directly from equation B2. The spatial derivatives of $\lambda$ in the case of prolate or oblate ellipsoids can also be calculated by using Eq. B22 for the particular case in with $b = c$.

*Author contributions.*  TEXT

*Acknowledgements.*  D. T. Tomazella thanks the brazilian research founding agency Conselho Nacional de Desenvolvimento Científico e Tecnológico (CNPq) for providing financial support in the form of a scholarship. Oliveira Jr. thanks the brazilian research founding agency Conselho Nacional de Desenvolvimento Científico e Tecnológico (CNPq) for providing financial support in the form of a grant (445752/2014-9).



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



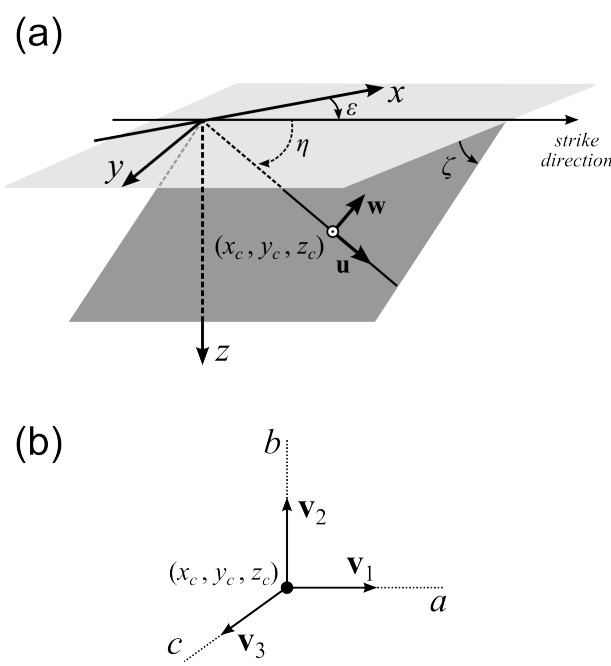

**Figure 1.** Schematic representation of the coordinate systems used to represent an ellipsoidal body. (a) Main coordinate system with axes $x$, $y$, and $z$ pointing to North, East, ans down, respectively. The dark gray plane contains the centre $(x_c, y_c, z_c)$ (white circle) and two unit vectors, $\mathbf{u}$ and $\mathbf{w}$, defining two semi-axes of the ellipsoidal body. For triaxial and prolate ellipsoids, $\mathbf{u}$ and $\mathbf{w}$ define, respectively, the semi-axes $a$ and $b$. For oblate ellipsoids, $\mathbf{u}$ and $\mathbf{w}$ define the semi-axes $b$ and $c$, respectively. The strike direction is defined by the intersection of the dark gray plane and the horizontal plane (represented in light gray), which contains the $x$ and $y$ axes. The angle $\varepsilon$ between the $x$-axis and the strike direction is called *strike*. The angle $\zeta$ between the horizontal plane and the dark gray plane is called *dip*. The angle $\eta$ between the strike direction and the line containing the unit vector $\mathbf{u}$ is called *rake*. The projection of this line on the horizontal plane (not shown) is called *dip direction* (Pollard and Fletcher, 2005; Allmendinger et al., 2012). (b) Local coordinate system with origin at the ellipsoid centre $(x_c, y_c, z_c)$ (black dot) and axes defined by unit vectors $\mathbf{v}_1$, $\mathbf{v}_2$, and $\mathbf{v}_3$. These unit vectors define the semi-axes $a$, $b$, and $c$ of triaxial, prolate, and oblate ellipsoids in the same way. For triaxial and prolate ellipsoids, the unit vectors $\mathbf{u}$ and $\mathbf{w}$ shown in (a) coincide with $\mathbf{v}_1$ and $\mathbf{v}_2$, respectively. For oblate ellipsoids, the unit vectors $\mathbf{u}$ and $\mathbf{w}$ shown in (a) coincide with $\mathbf{v}_2$ and $\mathbf{v}_3$, respectively.





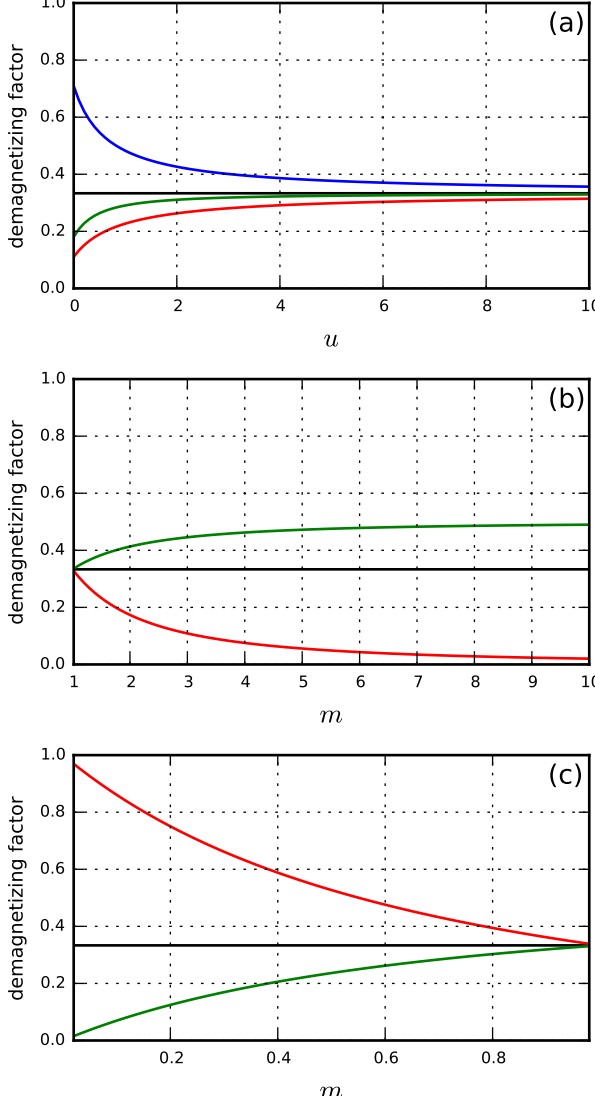

**Figure 2.** (a) Comparison between the demagnetizing factors $\tilde{n}^{\dagger}_{11}$ (in red), $\tilde{n}^{\dagger}_{22}$ (in green), and $\tilde{n}^{\dagger}_{33}$ (in blue) produced by 100 triaxial ellipsoids with semi-axes $a = a_0 + u\,b_0$, $b = b_0 + u\,b_0$, and $c = c_0 + u\,b_0$, where $0 \leq u \leq 10$ and $b_0 = 700$ m. The demagnetizing factors were calculated by using Eqs. 25, 26, and 27. (b) Comparison between the demagnetizing factors $\tilde{n}^{\dagger}_{11}$ (in red) and $\tilde{n}^{\dagger}_{22}$ (in green) produced by 100 prolate ellipsoids with semi-axes $a = m\,b_0$ and $b = b_0$, where $1.02 \leq m \leq 10$ and $b_0 = 1000$ m. The demagnetizing factors were calculated by using Eqs. 30 and 31. (c) Comparison between the demagnetizing factors $\tilde{n}^{\dagger}_{11}$ (in red) and $\tilde{n}^{\dagger}_{22}$ (in green) produced by 100 oblate ellipsoids with semi-axes $a = m\,b_0$ and $b = b_0$, where $0.02 \leq m \leq 0.98$ and $b_0 = 1000$ m. The demagnetizing factors were calculated by using Eqs. 32 and 33. The horizontal black line represent the value $1/3$.



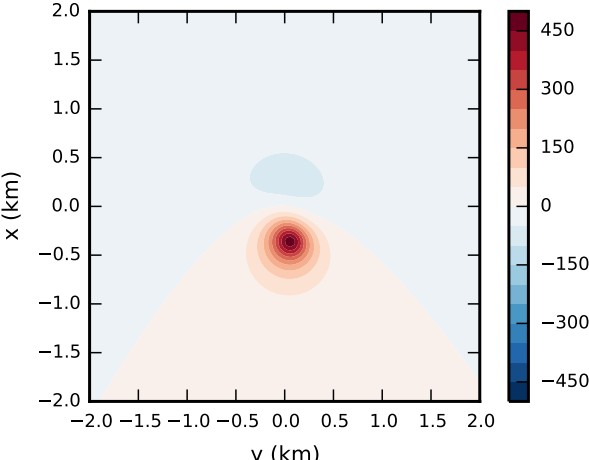

**Figure 3.** Total-field anomaly produced by the synthetic orebody defined in Tab. 1. The synthetic data are calculated on a regular grid of $100 \times 100$ points at the constant vertical coordinate $z = 0$ m.

**Table 1.** Parameters defining a synthetic orebody. This model is based on the that presented by Farrar (1979) to simulate the Warrego orebody, Tennant Creek field, Australia.

| Parameter | Value | Unit |
|---|---|---|
| Semi-axis $a$ | 490.7 | m |
| Semi-axis $b$ | 69.7 | m |
| Semi-axis $c$ | 30.0 | m |
| Coordinate of the centre $x_c$ | 0 | m |
| Coordinate of the centre $y_c$ | 0 | m |
| Coordinate of the centre $z_c$ | 500 | m |
| Orientation angle $\varepsilon$ * | −34.0 | ° |
| Orientation angle $\zeta$ * | 66.1 | ° |
| Orientation angle $\eta$ * | 45.0 | ° |
| Isotropic susceptibility $\chi$ | 1.69 | SI |
| $x$-component of the inducing field $\mathbf{B}_0$ † | 32610 | nT |
| $y$-component of the inducing field $\mathbf{B}_0$ † | 0 | nT |
| $z$-component of the inducing field $\mathbf{B}_0$ † | 39450 | nT |

* Defined in Fig. 1a

† Defined in Eq. 62



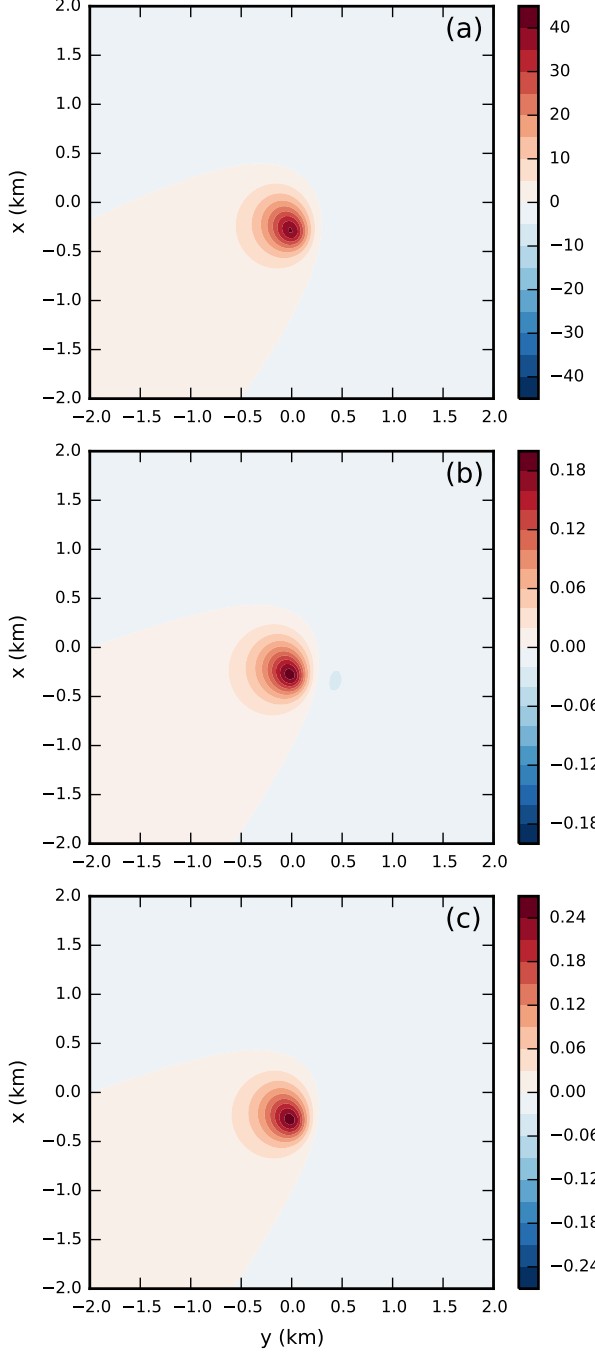

**Figure 4.** Difference between the total-field anomaly calculated with the approximated magnetization $\breve{\mathbf{M}}$ (Eq. 51) and with the true magneti-zation $\mathbf{M}$ (Eqs. 49 and 52). The total-field anomalies are calculated with Eq. 63, on a regular grid of $100 \times 100$ points, at the constant vertical coordinate $z = 0$ m. The differences are produced by the synthetic orebody defined in Tab. 1, but with different isotropic susceptibilities: (a) the isotropic susceptibility defined in Tab. 1, (b) an isotropic susceptibility $\chi = 0.1$ SI, and (c) an isotropic susceptibility $\chi = 0.116$ SI. This last value was calculated with Eq. 60, by using $\epsilon = 8\%$.