# Peer review of "Ellipsoids (v1.0): 3D Magnetic modelling of ellipsoidal bodies"

_Geoscientific Model Development, 2017_

## Referee Comment (RC1) · D. Clark (Referee) · 29 May 2017

This is a well-written exposition of the theory that underlies ellipsoid modelling, with practical algorithms. The mathematics, apart from a few tweaks, is not original, but it is clearly presented in a manner that should assist researchers who want to make use of ellipsoid modelling. The authors' new criterion for assessing the maximum susceptibilities for which self-demagnetization can be neglected is a useful addition to the theory. The software should prove very useful for the geophysical community. I have a few specific comments. 1. On line 5 "only finite bodies" might be better than "only bodies". 2. Line 27. The geoscientific community does indeed lack a FREE easy-to-use-tool for ellipsoid modelling, but commercial software packages (which are not cheap), such as

Tensor Research's ModelVision or GSS Potent, include ellipsoid modelling. 3. I think it is worth pointing out to the readers that there is a fundamental non-uniqueness of ellipsoidal sources, analogous to the equivalence of concentric spheres with the same magnetic moment. As pointed out by Clark (2014), co-located confocal ellipsoids with the same total magnetic moment vector produce identical anomalies. As the size of the equivalent ellipsoid increases, while maintaining the positions of the foci, its eccentricity decreases. Note that this does not imply, for isotropic susceptibility without remanence, that confocal ellipsoids with appropriately scaled susceptibilities will produce identical anomalies, unless the geomagnetic field happens to lie along one of the principal axes. If the field is oblique to the axes, then the deflection of the induced magnetization due to shape anisotropy will vary, depending on the size of the ellipsoid. In practice, the presence of remanence or anisotropy introduces uncertainty into interpretation of the orientation and axial ratios of the ellipsoid from its total magnetic moment. 4. It would be a simple matter to include remanence into the model. I recommend this for a future version. 5. Perhaps a future version could also generalize the model to consider ellipsoids immersed in a permeable medium. Stratton (2007) gives formulas that could be used for this.

---

## Author Comment (AC1) · 10 Jun 2017

Dear Dr. Clark,

First, we would like to thank you for reading and reviewing our manuscript, as well as thank you for all the constructive comments made in this discussion.

Indeed, there is no original mathematical breakthrough about ellipsoid modelling in our manuscript, but a review of a disperse literature that has been published since the

second half of the nineteenth century.

The response for the specific comments:

1 - *On line 5 "only finite bodies" might be better than "only bodies".*

Thank you, we have included the word "finite".

2 - *Line 27. The geoscientific community does indeed lack a FREE easy-to-use-tool for ellipsoid modelling, but commercial software packages (which are not cheap), such as Tensor Research's ModelVision or GSS Potent, include ellipsoid modelling.*

Thank you for your review. We have included a sentence pointing out the absence of free softwares for the magnetic modelling of ellipsoidal bodies.

3 - *I think it is worth pointing out to the readers that there is a fundamental non-uniqueness of ellipsoidal sources, analogous to the equivalence of concentric spheres with the same magnetic moment. As pointed out by Clark (2014), co-located confocal ellipsoids with the same total magnetic moment vector produce identical anomalies. As the size of the equivalent ellipsoid increases, while maintaining the positions of the foci, its eccentricity decreases. Note that this does not imply, for isotropic susceptibility without remanence, that confocal ellipsoids with appropriately scaled susceptibilities will produce identical anomalies, unless the geomagnetic field happens to lie along one of the principal axes. If the field is oblique to the axes, then the deflection of the induced magnetization due to shape anisotropy will vary, depending on the size of the ellipsoid. In practice, the presence of remanence or anisotropy introduces uncertainty into interpretation of the orientation and axial ratios of the ellipsoid from its total magnetic moment.*

This is a very good suggestion. We have included a new section presenting a discussion about this ambiguity and also included a numerical simulation illustrating its effect.

The code for generating the results presented in this new section can be found at:

http://nbviewer.jupyter.org/github/pinga-lab/magnetic-ellipsoid/blob/master/code/confocal_triaxial_ellipsoids.ipynb

4 - *It would be a simple matter to include remanence into the model. I recommend this for a future version.*

Actually, the remanence is already included in our code. Although we have not included in the manuscript a numerical test illustrating an ellipsoid with remanence, we have created some cookbooks that can be found at the online repository:

http://nbviewer.jupyter.org/github/pinga-lab/magnetic-ellipsoid/blob/master/code/Cookbook_triaxial.ipynb

http://nbviewer.jupyter.org/github/pinga-lab/magnetic-ellipsoid/blob/master/code/Cookbook_prolate.ipynb

http://nbviewer.jupyter.org/github/pinga-lab/magnetic-ellipsoid/blob/master/code/Cookbook_oblate.ipynb

These notebooks illustrate the field produced by triaxial, prolate and oblate ellipsoids with remanence. We could include, in the manuscript, a section showing these cookbooks.

5 - *Perhaps a future version could also generalize the model to consider ellipsoids immersed in a permeable medium. Stratton (2007) gives formulas that could be used for this.*

Thank you for your suggestion. We will consider this in a future version of our code.

Please also note the supplement to this comment:
http://www.geosci-model-dev-discuss.net/gmd-2017-44/gmd-2017-44-AC1-supplement.pdf

[Figure]

**Supplement:**

[revised manuscript text omitted]
{array}{ccc} x & y & z \end{array}]^\top$, $\mathbf{r}_c = [\begin{array}{ccc} x_c & y_c & z_c \end{array}]^\top$, $\mathbf{A}$ is a positive definite matrix given by

$$\mathbf{A} = \mathbf{V} \begin{bmatrix} a^{-2} & 0 & 0 \\ 0 & b^{-2} & 0 \\ 0 & 0 & c^{-2} \end{bmatrix} \mathbf{V}^\top, \tag{2}$$

15 and $\mathbf{V}$ is an orthogonal matrix whose first, second and third columns are defined by unit vectors $\mathbf{v}_1$, $\mathbf{v}_2$, and $\mathbf{v}_3$ (Fig. 1b), respectively.  The matrix $\mathbf{V}$ can be defined  in terms of three rotation matrices:

$$\alpha \mathbf{R}_1(\theta) = \varepsilon - \cos^{-1} \begin{bmatrix} \dfrac{\cos \eta}{\left(1 - \sin^2 \zeta \sin^2 \eta\right)^{\frac{1}{2}}} & 1 & 0 & 0 \\ & 0 & \cos \theta & \sin \theta \\ & 0 & -\sin \theta & \cos \theta \end{bmatrix}, \tag{3}$$

$$\gamma \mathbf{R}_2(\theta) = \tan^{-1} \dfrac{\cos \zeta}{\sin \zeta \cos \eta} \begin{bmatrix} \cos \theta & 0 & -\sin \theta \\ 0 & 1 & 0 \\ \sin \theta & 0 & \cos \theta \end{bmatrix} \tag{4}$$

and

$$\delta \mathbf{R}_3(\theta) = \sin^{-1} \sin \zeta \sin \eta \begin{bmatrix} \cos \theta & \sin \theta & 0 \\ -\sin \theta & \cos \theta & 0 \\ 0 & 0 & 1 \end{bmatrix} ,. \tag{5}$$

where $-90° \leq \gamma \leq 90°$ and $0 \leq \delta \leq 90°$. Then, given the orientation angles $\varepsilon, \zeta$, and $\eta$ (Fig. 1a), we define the auxiliary angles $\alpha, \gamma$ and $\delta$ (Eqs. ??, ??, and ??) and, finally, the unit vectors $\mathbf{v}_1$, $\mathbf{v}_2$, and $\mathbf{v}_3$ (Fig. 1b) according to the ellipsoid type. For triaxial ellipsoids (i.e., $a > b > c$) and prolate ellipsoids (i.e., $a > b = c$), we opted for defining these unit vectors by following Clark et al. (1986):

$$\mathbf{v}_1 = \begin{bmatrix} -\cos\alpha \ \cos\delta \\ -\sin\alpha \ \cos\delta \\ -\sin\delta \end{bmatrix},$$

$$\mathbf{v}_2 = \begin{bmatrix} \cos\alpha \ \cos\gamma \ \sin\delta + \sin\alpha \ \sin\gamma \\ \sin\alpha \ \cos\gamma \ \sin\delta - \cos\alpha \ \sin\gamma \\ -\cos\gamma \ \cos\delta \end{bmatrix},$$

$$\mathbf{v}_3 = \begin{bmatrix} \sin\alpha \ \cos\gamma - \cos\alpha \ \sin\gamma \ \sin\delta \\ -\cos\alpha \ \cos\gamma - \sin\alpha \ \sin\gamma \ \sin\delta \\ \sin\gamma \ \cos\delta \end{bmatrix}.$$

10 Emerson et al. (1985) calculated the unit vectors $\mathbf{v}_1$, $\mathbf{v}_2$, and $\mathbf{v}_3$ similarly by using Eqs. ??, ??, and ??, but with $\gamma = 0°$. Finally, we define the unit vectors $\mathbf{v}_1$, $\mathbf{v}_2$, and $\mathbf{v}_3$ for define the orthogonal matrix $\mathbf{V}$ as follows:

$$\mathbf{V} = \mathbf{R}_1\left(\frac{\pi}{2}\right) \mathbf{R}_2\left(\varepsilon\right) \mathbf{R}_1\left(\frac{\pi}{2} - \zeta\right) \mathbf{R}_3\left(\eta\right). \tag{6}$$

For oblate ellipsoids (i.e., $a < b = c$), we define $\mathbf{V}$ as follows:

$$\mathbf{v}_1 \mathbf{V} = \mathbf{R}_3\left(-\frac{\pi}{2}\right) \mathbf{R}_1\left(\pi\right) \mathbf{R}_3\left(\varepsilon\right) \mathbf{R}_2\left(\frac{\pi}{2} - \zeta\right) \mathbf{R}_1\left(\eta\right). \tag{7}$$

$$\mathbf{v}_2 = \begin{bmatrix} -\cos\alpha \ \cos\delta \\ -\sin\alpha \ \cos\delta \\ -\sin\delta \end{bmatrix},$$

$$\mathbf{v}_3 = \begin{bmatrix} \sin\alpha \ \sin\gamma + \cos\alpha \ \cos\gamma \ \sin\delta \\ -\cos\alpha \ \sin\gamma + \sin\alpha \ \cos\gamma \ \sin\delta \\ -\cos\gamma \ \cos\delta \end{bmatrix}.$$

This approach is very similar to that presented by Emerson et al. (1985) The orthogonal matrices $\mathbf{V}$ used here for triaxial,

20 prolate and oblate ellipsoids (Eqs. 6 and 7) are different from that used by Emerson et al. (1985) and Clark et al. (1986).

The magnetic modelling of an ellipsoidal body is commonly performed in a particular Cartesian coordinate system that is aligned with the body semi-axes and has the origin coincident with the body centre (Fig. 1b). For convenience, we denominate

this particular coordinate system as *local coordinate system*. The relationship between the Cartesian coordinates $(\tilde{x}, \tilde{y}, \tilde{z})$ of a point in a local coordinate system and the Cartesian coordinates $(x, y, z)$ of the same point in the main system is given by:

$$\tilde{\mathbf{r}} = \mathbf{V}^{\top} (\mathbf{r} - \mathbf{r}_c) \,, \tag{8}$$

[revised manuscript text omitted]
\left(\kappa, \phi\right) - \frac{a^2 - b^2}{\left(a^2 - c^2\right)^{\frac{1}{2}}} \left[\frac{c^2 + \lambda}{\left(a^2 + \lambda\right)\left(b^2 + \lambda\right)}\right]^{\frac{1}{2}} \right\} \tag{39}$$

and

$$g_3 = \frac{2}{\left(b^2 - c^2\right)\left(a^2 - c^2\right)^{\frac{1}{2}}} E\left(\kappa, \phi\right) + \frac{2}{b^2 - c^2} \left[\frac{b^2 + \lambda}{\left(a^2 + \lambda\right)\left(c^2 + \lambda\right)}\right]^{\frac{1}{2}}, \tag{40}$$

where $F(\kappa, \phi)$ and $E(\kappa, \phi)$ are defined by Eqs. 29 and 28, but with $\sin\phi = \sqrt{\left(a^2 - c^2\right)/\left(a^2 + \lambda\right)}$. A detailed deduction of these formulas was presented by Tejedor et al. (1995). Similar formulas can also be found in Clark et al. (1986).

**Prolate ellipsoids**

For prolate (e.g., $a > b = c$) ellipsoids, the functions $g_i$ (Eq. 37) are given by:

$$g_1 = \frac{2}{\left(a^2 - b^2\right)^{\frac{3}{2}}} \left\{ \ln\left[\frac{\left(a^2 - b^2\right)^{\frac{1}{2}} + \left(a^2 + \lambda\right)^{\frac{1}{2}}}{\left(b^2 + \lambda\right)^{\frac{1}{2}}}\right] - \left(\frac{a^2 - b^2}{a^2 + \lambda}\right)^{\frac{1}{2}} \right\} \tag{41}$$

and

$$g_2 = \frac{1}{\left(a^2 - b^2\right)^{\frac{3}{2}}} \left\{ \frac{\left[\left(a^2 - b^2\right)\left(a^2 + \lambda\right)\right]^{\frac{1}{2}}}{b^2 + \lambda} - \ln\left[\frac{\left(a^2 - b^2\right)^{\frac{1}{2}} + \left(a^2 + \lambda\right)^{\frac{1}{2}}}{\left(b^2 + \lambda\right)^{\
[revised manuscript text omitted]

---

## Editor Comment (EC1) · L. Gross (Editor) · 6 Jul 2017

Diego

In order to maintain a high degree of reproducibility of results presented in papers GMD is strongly encouraging (but at this point does not enforce) authors to provide persistent access to their program code and data used in the manuscript. Typically this is guaranteed through the use of a DOI which can be created for releases made in GitHub using Zenodo, see https://guides.github.com/activities/citable-code/ for details. We are aware that this is not always possible or reasonable but looking at the weak dependencies in your program code (as far as I am able to identify them) I would like

to suggest that you look into the possibility to define a DOI for the version forming the basis of the manuscript results and cite this is the paper. Feel free to mentioned that the code is still being improved and you encourage the user to work with the newest version.

Thanks.

Lutz Gross GMD Executive Officer

---

## Referee Comment (RC2) · R. Schaa (Referee) · 8 Jul 2017

I could not find any flaws in this paper, it is diligently prepared with a keen eye for details and clearly presented; it was a pleasure to read. Dr Clark already addressed minor corrections and avenues for further development. I recommend the paper for publication.

This paper presents a review of the existing literature and mathematical descriptions of magnetic modelling of ellipsoids. I consider this a relevant scientific contribution as it integrates key points of the many different published papers on the matter into one paper. Equations for calculating demagnetization factors for ellipsoidal bodies are provided in

one consistent framework. Furthermore the 0.1 SI threshold for self-demagnetisation is here, apparently for the first time, analytically quantified and a practical formula is provided to estimate the threshold level as a function of a maximum demagnetization factor. The various equations have been integrated into the 'Fatiando a Terra' open source modelling package which provides the opportunity for researchers to replicate and validate the results shown.

One small nit-pick: p.6, ln 24: "which supported the Maxwell's (1873) postulate" – delete "the"

Ralf Schaa Curtin University
* * *

---

## Author Comment (AC3) · 10 Jul 2017

Dear Dr. Ralf Schaa,

Thank you very much for reviewing our manuscript and also for your kind comments. We have followed your suggestion and removed the "the" on page 6. We hope that our code be a useful tool for many researchers and students. Please, find a pdf containing the current version of our manuscript at the response that we have just written for the Editor.

Sincerely, Vanderlei

---

## Author Response (AR1)

Dear editor,

Thank you for your support. The recommendations from you and the reviewers have been very helpful to improve our manuscript. We hope you find that we have addressed all of the comments and recommendations appropriately and that the manuscript is in better shape for publication. A marked-up manuscript version showing the changes made is presented in the following pages.

Sincerely,

Vanderlei C. Oliveira Jr and Diego T. Tomazella

[revised manuscript text omitted]

---

## Author Response (AR2)

Dear Editor,

We kindly thank you for all your support and revision of our manuscript. We have included the reference to our code in the section "6 – Code Availability". We are sending the files for upload.

Kind regards,

Diego Tomazella and Vanderlei Oliveira Jr.